# Universal Physics Transformers: A Framework For Efficiently Scaling Neural Operators

**Benedikt Alkin** [1,2]    **Andreas Fürst** [1]    **Simon Schmid** [3]    **Lukas Gruber** [1]
**Markus Holzleitner** [4]    **Johannes Brandstetter** [1,2]

[1] ELLIS Unit Linz, Institute for Machine Learning, JKU Linz, Austria
[2] NXAI GmbH, Linz, Austria
[3] Software Competence Center Hagenberg GmbH, Hagenberg, Austria
[4] MaLGa Center, Department of Mathematics, University of Genoa, Italy
`{alkin, fuerst, brandstetter}@ml.jku.at`

## Abstract

Neural operators, serving as physics surrogate models, have recently gained increased interest. With ever increasing problem complexity, the natural question arises: what is an efficient way to scale neural operators to larger and more complex simulations – most importantly by taking into account different types of simulation datasets. This is of special interest since, akin to their numerical counterparts, different techniques are used across applications, even if the underlying dynamics of the systems are similar. Whereas the flexibility of transformers has enabled unified architectures across domains, neural operators mostly follow a problem specific design, where GNNs are commonly used for Lagrangian simulations and grid-based models predominate Eulerian simulations. We introduce Universal Physics Transformers (UPTs), an efficient and unified learning paradigm for a wide range of spatio-temporal problems. UPTs operate without grid- or particle-based latent structures, enabling flexibility and scalability across meshes and particles. UPTs efficiently propagate dynamics in the latent space, emphasized by inverse encoding and decoding techniques. Finally, UPTs allow for queries of the latent space representation at any point in space-time. We demonstrate diverse applicability and efficacy of UPTs in mesh-based fluid simulations, and steady-state Reynolds averaged Navier-Stokes simulations, and Lagrangian-based dynamics. Project page: `https://ml-jku.github.io/UPT`

## 1   Introduction

In scientific pursuits, extensive efforts have produced highly intricate mathematical models of physical phenomena, many of which are naturally expressed as partial differential equations (PDEs) [76]. Solving most PDEs is analytically intractable and necessitates falling back on compute-expensive numerical approximation schemes. In recent years, deep neural network based surrogates, most importantly neural operators [51, 61, 47], have emerged as a computationally efficient alternative [99, 119], and impact e.g., weather forecasting [48, 8, 1], molecular modeling [7, 4], or computational fluid dynamics [107, 30, 51, 43, 31, 19]. Additional to computational efficiency, neural surrogates offer potential to introduce generalization capabilities across phenomena, as well as generalization across characteristics such as boundary conditions or PDE coefficients [66, 14]. Consequently, the nature of neural operators inherently complements handcrafted numerical solvers which are characterized by a substantial set of solver requirements, and mostly due to these requirements tend to differ among sub-problems [3].

38th Conference on Neural Information Processing Systems (NeurIPS 2024).

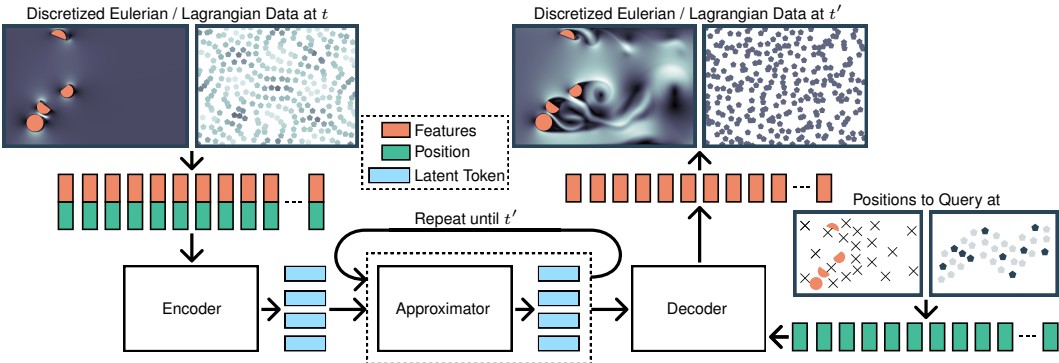

Figure 1: Schematic sketch of the UPT learning paradigm. UPTs flexible encode different grids, and/or different number of particles into a unified latent space representation, and subsequently unroll dynamics in the latent space. The latent space is kept at a fixed size to ensure scalability to larger systems. UPTs decode the latent representation at any query point.

However, similar to their numerical counterparts, different neural network techniques are prevalent across applications. For example, when contrasting particle- and grid-based dynamics in computational fluid dynamics (CFD), i.e., Lagrangian and Eulerian discretization schemes. This is in contrast to other areas of deep learning where the flexibility of transformers [106] has enabled unified architectures across domains, allowing advancements in one domain to also benefit all others. This has lead to an efficient scaling of architectures, paving the way for large "foundation" models [10] that are pretrained on huge passive datasets [25, 36].

We introduce Universal Physics Transformers (UPTs), an efficient and unified neural operator learning paradigm with strong focus on scalability over a wide range of spatio-temporal problems. UPTs flexibly encode different grids, and/or different number of particles into a compressed latent space representation which facilitates scaling to large-scale simulations. Latent space rollouts are enforced by inverse encoding and decoding surrogates, leading to fast simulated trajectories which is particularly important for large systems. For decoding, the latent representation can be evaluated at any point in space-time. UPTs operate without grid- or particle-based latent structures, and demonstrate the beneficial scaling-behavior of transformer backbone architectures. Figure 1 sketches the UPT modeling paradigm.

We summarize our contributions as follows: (i) we introduce the UPT framework for efficiently scaling neural operators; (ii) we formulate encoding and decoding schemes such that dynamics can be propagated efficiently in a compressed and fixed-size latent space; (iii) we demonstrate applicability on diverse applications, putting a strong research focus on the scalability of UPTs.

## 2  Background

**Partial differential equations**.    We focus on experiments on (systems of) PDEs, that evolve a signal $\boldsymbol{u}(t, \boldsymbol{x}) = \boldsymbol{u}^t(\boldsymbol{x}) \in \mathbb{R}^d$ in a single temporal dimension $t \in [0, T]$ and $m$ spatial dimensions $\boldsymbol{x} \in U \subset \mathbb{R}^m$, for an open set $U$. With $1 \leq l \in \mathbb{N}$, systems of PDEs of order $l$ can be written as

$$\boldsymbol{F}(D^l \boldsymbol{u}^t(\boldsymbol{x}), \ldots, D^1 \boldsymbol{u}^t(\boldsymbol{x}), \frac{\partial^l}{\partial t^l} \boldsymbol{u}^t(\boldsymbol{x}), \ldots, \frac{\partial}{\partial t} \boldsymbol{u}^t(\boldsymbol{x}), \boldsymbol{u}^t(\boldsymbol{x}), \boldsymbol{x}, t) = \boldsymbol{0}, \quad \text{for } \boldsymbol{x} \in U, t \in [0, T] \,,$$

(1)

where $\boldsymbol{F}$ is a mapping to $\mathbb{R}^d$, and for $i = 1, ..., l$, $D^i$ denotes the differential operator mapping to all $i$-th order partial derivatives of $\boldsymbol{u}$ with respect to the spacial variable $\boldsymbol{x}$, whereas $\frac{\partial^i}{\partial t^i}$ outputs the corresponding time derivative of order $i$. Any $l$-times continuously differentiable $\boldsymbol{u} : [0, T] \times U \to \mathbb{R}^d$ fulfilling the relation Eq. (1) is called a *classical solution* of Eq. (1). Also other notions of solvability (e.g. in the sense of weak derivatives/distributions) are possible, for sake of simplicity we do not go into details here. Additionally, initial conditions specify $\boldsymbol{u}^t(\boldsymbol{x})$ at time $t = 0$ and boundary conditions $B[\boldsymbol{u}^t](\boldsymbol{x})$ at the boundary of the spatial domain.

We work mostly with the incompressible Navier-Stokes equations [98], which e.g., in two spatial dimensions conserve the velocity flow field $\boldsymbol{u}(t, x, y) : [0, T] \times \mathbb{R}^2 \to \mathbb{R}^2$ via:

$$\frac{\partial \boldsymbol{u}}{\partial t} = -\boldsymbol{u} \cdot \nabla \boldsymbol{u} + \mu \nabla^2 \boldsymbol{u} - \nabla p + \boldsymbol{f} \,, \quad \nabla \cdot \boldsymbol{u} = 0 \,, \tag{2}$$

where $\boldsymbol{u} \cdot \nabla \boldsymbol{u}$ is the convection, i.e., the rate of change of $\boldsymbol{u}$ along $\boldsymbol{u}$, $\mu$ is the viscosity parameter, $\mu \nabla^2 \boldsymbol{u}$ the viscosity, i.e., the diffusion or net movement of $\boldsymbol{u}$, $\nabla p$ the internal pressure gradient, and $\boldsymbol{f}$ an external force. The constraint $\nabla \cdot \boldsymbol{u} = 0$ yields mass conservation of the Navier-Stokes equations. A detailed depiction of the involved differential operators is given in Appendix B.1.

**Operator learning.** Operator learning [60, 61, 52, 51, 47] learns a mapping between function spaces – a concept which is often used to approximate solutions of PDEs. Similar to Kovachki et al. [47], we assume $\mathcal{U}, \mathcal{V}$ to be Banach spaces of functions on compact domains $\mathcal{X} \subset \mathbb{R}^{d_x}$ or $\mathcal{Y} \subset \mathbb{R}^{d_y}$, mapping into $\mathbb{R}^{d_u}$ or $\mathbb{R}^{d_v}$, respectively. The goal of operator learning is to learn a ground truth operator $\mathcal{G} : \mathcal{U} \to \mathcal{V}$ via an approximation $\hat{\mathcal{G}} : \mathcal{U} \to \mathcal{V}$. This is usually done in the vein of supervised learning by i.i.d. sampling input-output pairs, with the notable difference, that in operator learning the spaces sampled from are not finite dimensional. More precisely, with a given data set consisting of $N$ function pairs $(\boldsymbol{u}_i, \boldsymbol{v}_i) = (\boldsymbol{u}_i, \mathcal{G}(\boldsymbol{u}_i)) \subset \mathcal{U} \times \mathcal{V}$, $i = 1, ...N$, we aim to learn $\hat{\mathcal{G}} : \mathcal{U} \to \mathcal{V}$, so that $\mathcal{G}$ can be approximated in a suitably chosen norm.

In the context of PDEs, $\mathcal{G}$ can e.g. be the mapping from an initial condition $\boldsymbol{u}(0, \boldsymbol{x}) = \boldsymbol{u}^0(\boldsymbol{x})$ to the solutions $\boldsymbol{u}(t, \boldsymbol{x}) = \boldsymbol{u}^t(\boldsymbol{x})$ of Eq. (1) at all times. In the case of classical solutions, if $U$ is bounded, $\mathcal{U}$ can then be chosen as a subspace of $C(\bar{U}, \mathbb{R}^d)$, the set of continuous functions from domain $\bar{U}$ (the closure of $U$) mapping to $\mathbb{R}^d$, whereas $\mathcal{V} \subset C([0, T] \times \bar{U}, \mathbb{R}^d)$, so that $\mathcal{U}$ or $\mathcal{V}$ consist of all $l$-times continuosly differentiable functions on the respective spaces. In case of weak solutions, the associated spaces $\mathcal{U}$ and $\mathcal{V}$ can be chosen as Sobolev spaces.

We follow the popular approach to approximate $\mathcal{G}$ via three maps [89]: $\mathcal{G} \approx \hat{\mathcal{G}} := \mathcal{D} \circ \mathcal{A} \circ \mathcal{E}$. The encoder $\mathcal{E} : \mathcal{U} \to \mathbb{R}^{h_1}$ takes an input function and maps it to a finite dimensional latent feature representation. For example, $\mathcal{E}$ could embed a continuous function to a chosen hidden dimension $\mathbb{R}^{h_1}$ for a collection of grid points. Next, $\mathcal{A} : \mathbb{R}^{h_1} \to \mathbb{R}^{h_2}$ approximates the action of the operator $\mathcal{G}$, and $\mathcal{D}$ decodes the hidden representation, and thus creates the output functions via $\mathcal{D} : \mathbb{R}^{h_2} \to \mathcal{V}$, which in many cases is point-wise evaluated at the output grid or output mesh.

**Particle vs. grid-based methods.** Often, numerical simulation methods can be classified into two distinct families: particle and grid-based methods. This specification is notably prevalent, for instance, in the field of computational fluid dynamics (CFD), where Lagrangian and Eulerian discretization schemes offer different characteristics dependent on the PDEs. In simpler terms, Eulerian schemes essentially monitor velocities at specific fixed grid points. These points, represented by a spatially limited number of nodes, control volumes, or cells, serve to discretize the continuous space. This process leads to grid-based or mesh-based representations. In contrast to such grid- and mesh-based representations, in Lagrangian schemes, the discretization is carried out using finitely many material points, often referred to as particles, which move with the local deformation of the continuum. Roughly speaking, there are three families of Lagrangian schemes: discrete element methods [24], material point methods [94, 12], and smoothed particle hydrodynamics (SPH) [29, 62, 69, 70]. In this work, we focus on SPH methods, which approximate the field properties using radial kernel interpolations over adjacent particles at the location of each particle. The strength of SPH lies in its ability to operate without being constrained by connectivity issues, such as meshes. This characteristic proves especially beneficial when simulating systems that undergo significant deformations.

**Latent space representation of neural operators.** For larger meshes or larger number of particles, memory consumption and inference speed become more and more important. Fourier Neural Operator (FNO) based methods work on regular grids, or learn a mapping to a regular latent grid, e.g., geometry-informed neural operators (GINO) [54]. In three dimensions, the stored Fourier modes have the shape $h \times n_x \times n_y \times n_z$, where $h$ is the hidden size and $n_x, n_y, n_z$ are the respective Fourier modes. Similarly, the latent space of CNN-based methods, e.g., Raonić et al. [83], Gupta & Brandstetter [31], is of shape $h \times w_x \times w_y \times w_z$, where $w_x, w_y, w_z$ are the respective grid points. In three dimension, the memory requirement in each layer increases cubically with increasing number of modes or grid points. In contrast, transformer based neural operators, e.g., Hao et al. [33], Cao [18], Li et al. [53], operate on a token-based latent space of dimension $n_{\text{tokens}} \times h$, where usually $n_{\text{tokens}} \propto n_{\text{points}}$, and GNN based neural operators, e.g., Li et al. [52], operate on a node based latent space of dimension

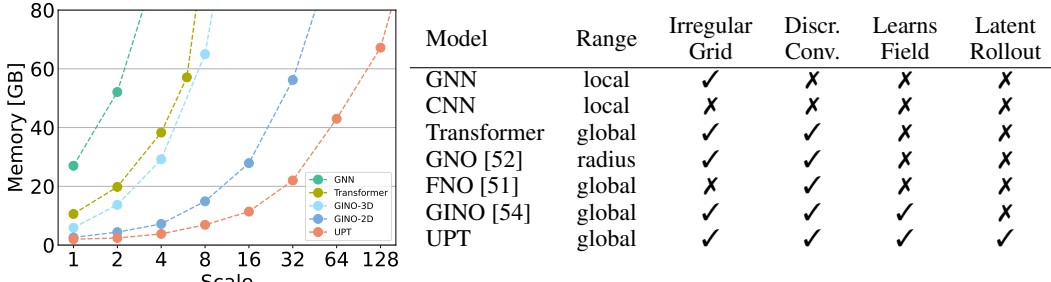

| Model | Range | Irregular Grid | Discr. Conv. | Learns Field | Latent Rollout |
|---|---|---|---|---|---|
| GNN | local | ✓ | ✗ | ✗ | ✗ |
| CNN | local | ✗ | ✗ | ✗ | ✗ |
| Transformer | global | ✓ | ✓ | ✗ | ✗ |
| GNO [52] | radius | ✓ | ✓ | ✗ | ✗ |
| FNO [51] | global | ✗ | ✓ | ✗ | ✗ |
| GINO [54] | global | ✓ | ✓ | ✓ | ✗ |
| UPT | global | ✓ | ✓ | ✓ | ✓ |

Figure 2: Qualitative exploration of scaling limits. Starting from 32K input points (scale 1), we train a 68M parameter model for a few steps with batchsize 1 and measure the required GPU memory. Models without a compressed latent space (GNN, Transformer) quickly reach their limits while models with a compressed latent space (GINO, UPT) scale much better with the number of inputs. However, as GINO compresses the latent space onto a regular grid, the scaling benefits are largely voided on 3D problems. The efficient latent space compression of UPTs can fit up to 4.2M points (scale 128). We use a linear attention transformer [90] for this study. "Disc. Conv." denotes "Discretization Convergent". Appendix D.7 outlines implementation details and complexities.

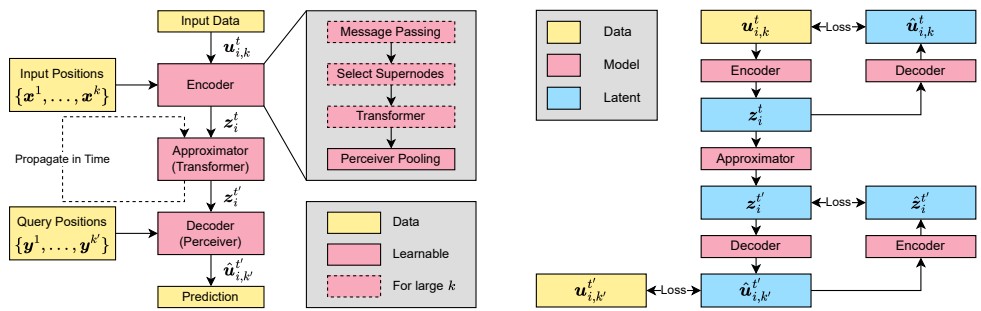

Figure 3: Left: UPT compresses information from various grids or differing particles with an encoder, propagates this information forward in time through the approximator and decodes at arbitrary query positions. Right: Training procedure to enable latent rollouts via inverse encoding/decoding losses.

$n_{\text{nodes}} \times h$, where usually $n_{\text{nodes}} = n_{\text{points}}$. For large number of inputs, this becomes infeasible as every layer has to process a large number of tokens. Contrary, UPTs compress the inputs into a low-dimensional latent space, which drastically decreases computational requirements. Different architectures and their scaling limits are compared in Fig. 2.

## 3 Universal Physics Transformers

**Problem formulation**. Our goal is to learn a mapping between the solutions $\boldsymbol{u}^t$ and $\boldsymbol{u}^{t'}$ of Eq. (1) at timesteps $t$ and $t'$, respectively. Our dataset should consist of $N$ function pairs $(\boldsymbol{u}_i^t, \boldsymbol{u}_i^{t'})$, $i = 1, .., N$, where each $\boldsymbol{u}_i^t$ is sampled at $k$ spatial locations $\{\boldsymbol{x}_i^1, \ldots, \boldsymbol{x}_i^k\} \in U$. Similarly, we query each output signal $\hat{\boldsymbol{u}}_i^{t'}$ at $k'$ spatial locations $\{\boldsymbol{y}_i^1, \ldots, \boldsymbol{y}_i^{k'}\} \in U$. Then each input signal can be represented by $\boldsymbol{u}_{i,k}^t = (\boldsymbol{u}_i^t(\boldsymbol{x}_i^1), \ldots, \boldsymbol{u}_i^t(\boldsymbol{x}_i^k))^T \in \mathbb{R}^{k \times d}$ as a tensor of shape $k \times d$, similar for the output. For particle- or mesh-based inputs, it is often simpler to represent the input as graph $\mathcal{G} = (V, E)$ with $k$ nodes $\{\boldsymbol{x}_i^1, \ldots \boldsymbol{x}_i^k\} \in V$, edges $E$ and node features $\{\boldsymbol{u}_i^t(\boldsymbol{x}_i^1), \ldots, \boldsymbol{u}_i^t(\boldsymbol{x}_i^k)\}$.

**Architecture desiderata**. We want Universal Physics Transformers (UPTs) to fulfill the following set of desiderata: (i) an encoder $\mathcal{E}$, which flexibly encodes different grids, and/or different number of particles into a unified latent representation of shape $n_{\text{latent}} \times h$, where $n_{\text{latent}}$ is the chosen number of tokens in the latent space and $h$ is the hidden dimension; (ii) an approximator $\mathcal{A}$ and a training procedure, which allows us to forward propagate dynamics purely within the latent space without mapping back to the spatial domain at each operator step; and iii) a decoder $\mathcal{D}$ that queries the latent representation at different locations. The UPT architecture is schematically sketched in Fig. 3.

**Encoder.** The goal of the encoder $\mathcal{E}$ is to compress the input signal $\boldsymbol{u}_i^t$, which is represented by a point cloud $\boldsymbol{u}_{i,k}^t$. Importantly, the encoder should learn to selectively focus on important parts of the input. This is a desirable property as, for example, in many computational fluid dynamics simulations large areas are characterized by laminar flows, whereas turbulent flows tend to occur especially around obstacles. If $k$ is large, we employ a hierarchical encoder.

The encoder $\mathcal{E}$ first embeds $k$ points into hidden dimension $h$, adding position encoding [106] to the different nodes, i.e., $\boldsymbol{u}_{i,k}^t \in \mathbb{R}^{k \times d} \to \mathbb{R}^{k \times h}$. In the first hierarchy, information is exchanged between local points and a selected set of $n_s$ supernode points. For Eulerian discretization schemes those supernodes can either be uniformly sampled on a regular grid as in [54], or selected based on the given mesh. The latter has the advantage that mesh characteristics are automatically taken into account, e.g., dense or sparse mesh regions are represented by different numbers of nodes. Furthermore, adaptation to new meshes is straightforward. We implement the first hierarchy by randomly selecting $n_s$ supernodes on the mesh, choosing $n_s$ such that the mesh characteristic is preserved. Similarly, in the Lagrangian discretization scheme, choosing supernodes based on particle positions provides the same advantages as selecting them based on the mesh.

Information is aggregated at the selected $n_s$ supernodes via a message passing layer [28] using a radius graph between points. Importantly, messages only flow towards the $n_s$ supernodes, and thus the compute complexity of the first hierarchy scales linearly with $n_s$. The second hierarchy consists of transformer blocks [106] followed by a perceiver block [40, 39] with $n_{\text{latent}}$ learned queries of dimension $h$. To summarize, the encoder $\mathcal{E}$ maps $\boldsymbol{u}_i^t \in \mathcal{U}$ to a latent space via

$$\mathcal{E} : \boldsymbol{u}_i^t \in \mathcal{U} \xrightarrow{\text{evaluate}} \boldsymbol{u}_{i,k}^t \in \mathbb{R}^{k \times d} \xrightarrow{\text{embed}} \mathbb{R}^{k \times h} \xrightarrow{\text{MP}} \mathbb{R}^{n_s \times h}$$

$$\xrightarrow{\text{transformer}} \mathbb{R}^{n_s \times h} \xrightarrow{\text{perceiver}} \boldsymbol{z}_i^t \in \mathbb{R}^{n_{\text{latent}} \times h} \ ,$$

where tyically $n_{\text{latent}} \ll n_s < k$. If the number of points is small, the first hierarchy can be omitted.

Note that randomly sampling mesh cells or particles implicitly encodes the underlying mesh or particle density and allocates more supernodes to highly resolved areas in the mesh or densely populated particle regions. Therefore, this can be seen as an implicit "importance sampling" of the underlying simulation. Additional implementation details are provided in Appendix F.1.

This encoder design projects into a fixed size latent space as it is an efficient way to compress the input into a fixed size representation to enable scaling to large-scale systems while remaining compute efficient. However, if an application requires a variable sized latent space, one could also remove the perceiver pooling layer. With this change the number of supernodes is equal to the number of latent tokens and complex problems could be tackled by a larger supernode count.

**Approximator.** The approximator propagates the compressed representation forward in time. As $n_{\text{latent}}$ is small, forward propagation in time is fast. We employ a transformer as approximator.

$$\mathcal{A} : \boldsymbol{z}_i^t \in \mathbb{R}^{n_{\text{latent}} \times h} \to \boldsymbol{z}_i^{t'} \in \mathbb{R}^{n_{\text{latent}} \times h} \ .$$

Notably, the approximator can be applied multiple times, propagating the signal forward in time by $\Delta t$ each time. If $\Delta t$ is small enough, the input signal can be approximated at arbitrary future times $t'$.

**Decoder.** The task of decoder $\mathcal{D}$ is to query the latent representation at $k'$ arbitrary locations to construct the prediction of the output signal $\boldsymbol{u}_i^{t'}$ at time $t'$. More formally, given the output positions $\{\boldsymbol{y}_i^1, \ldots, \boldsymbol{y}_i^{k'}\} \in U$ at $k'$ spatial locations and the latent representation $\boldsymbol{z}_i^{t'}$, the decoder predicts the output signal $\boldsymbol{u}_{i,k'}^{t'} = (\boldsymbol{u}_i^{t'}(\boldsymbol{y}_i^1), \ldots, \boldsymbol{u}_i^{t'}(\boldsymbol{y}_i^{k'}))^T$ at these spatial locations at timestep $t'$,

$$\mathcal{D} : (\boldsymbol{z}_i^{t'}, \{\boldsymbol{y}_i^1, \ldots, \boldsymbol{y}_i^{k'}\}) \to \hat{\boldsymbol{u}}_{i,k'}^{t'} \in \mathbb{R}^{k' \times d} \ .$$

The decoder is implemented via a perceiver-like cross attention layer using a positional embedding of the output positions as query and the latent representation $\boldsymbol{z}_i^{t'}$ as keys and values. Since there is no interaction between queries, the latent representation can be queried at arbitrarily many positions without large computational overhead. This decoding mechanism establishes a connection of conditioned neural fields to operator learning [79].

**Model Conditioning.** To condition the model to the current timestep $t$ and to boundary conditions such as the inflow velocity, we add feature modulation to all transformer and perceiver blocks. We use DiT modulation [78], which consists of a dimension-wise scale, shift and gate operation that are

applied to the attention and MLP module of the transformer. Scale, shift and gate are dependent on an embedding of the timestep and boundary conditions (e.g. velocity).

**Training procedure**. UPTs model the dynamics fully within a latent representation, such that during inference only the initial state of the system $\boldsymbol{u}(0, \boldsymbol{x}) = \boldsymbol{u}^0(\boldsymbol{x})$ is encoded into a latent representation $\boldsymbol{z}^0$. From there on, instead of autoregressively feeding the decoder's prediction into the encoder, UPTs propagate $\boldsymbol{z}^0$ forward in time to $\boldsymbol{z}^{t'}$ through iteratively applying the approximator $\mathcal{A}$ in the latent space. We call this procedure *latent rollout*. Especially for large meshes or many particles, the benefits of latent space rollouts, i.e. fast inference, pays off.

To enable latent rollouts, the responsibilities of encoder $\mathcal{E}$, approximator $\mathcal{A}$ and decoder $\mathcal{D}$ need to be isolated. Therefore, we invert the encoding and decoding by means of two reconstruction losses during training as visualized in Fig. 3. First, an inverse encoding is performed, wherein the input $\boldsymbol{u}_i^t$ is reconstructed from the encoded latent state $\boldsymbol{z}_i^t$ by querying it with the decoder at $k$ input locations $\{\boldsymbol{x}_i^1, \dots, \boldsymbol{x}_i^k\}$. Second, we invert the decoding by reconstructing the latent state $\boldsymbol{z}_i^{t'}$ from the output signal $\hat{\boldsymbol{u}}_i^{t'}$ at $k'$ spatial locations $\{\boldsymbol{y}_i^1, \dots, \boldsymbol{y}_i^{k'}\}$. Using two reconstruction losses, the encoder is forced to focus on encoding a state $\boldsymbol{u}_i^t$ into a latent representation $\boldsymbol{z}^t$, and similarly the decoder is forced to focus on making predictions out of a latent representation $\boldsymbol{z}^{t'}$.

**Related methods**. The closest work to ours are transformer neural operators of Cao [18], Li et al. [53], Hao et al. [33] which encode different query points into a tokenized latent space representation of dimension $n_{\text{nodes}} \times h$, where $n_{\text{nodes}}$ varies based on the number of input points, i.e., $n_{\text{nodes}} \propto n_{\text{points}}$. Wu et al. [114] adds a learnable mapping into a fixed latent space of dimension $n_{\text{nodes}} \times h$ to each transformer layer, and projects back to dimension $n_{\text{points}} \times h$ after self-attention. In contrast, UPTs use fixed $n_{\text{latent}}$ for the unified latent space representation $n_{\text{latent}} \times h$.

For the modeling of temporal PDEs, a common scheme is to map the input solution at time $t$ to the solution at next time step $t'$ [51, 13, 95]. Especially for systems that are modeled by graph-based representations, predicted accelerations at nodes are numerically integrated to model the time evolution of the system [87, 81]. Recently, equivariant graph neural operators [116] were introduced which model time evolution via temporal convolutions in Fourier space. More related to our work are methods that propagate dynamics in the latent space [50, 111]. Once the system is encoded, time evolution is modeled via LSTMs [111], or even linear propagators [63, 73]. In Li et al. [53], attention-based layers are used for encoding the spatial information of the input and query points, while time updates in the latent space are performed using recurrent MLPs. Similarly, Bryutkin et al. [16] use recurrent MLPs for temporal updates within the latent space, while utilizing a graph transformer for encoding the input observations.

Building universal models aligns with the contemporary trend of foundation models for science. Recent works comprise pretraining over multiple heterogeneous physical systems, mostly in the form of PDEs [66], foundation models for weather and climate [75], or material modeling [67, 118, 4].

Methods similar to our latent space modeling have been proposed in the context of diffusion models [85] where a pre-trained compression model is used to compress the input into a latent space from which a diffusion model can be trained at much lower costs. Similarly, our approach also compresses the high-dimensional input into a low-dimensional latent space, but without a two stage approach. Instead, we learn the compression end-to-end via inverse encoding and decoding techniques.

## 4 Experiments

We ran experiments across different settings, assessing three key aspects of UPTs: (i) **Effectiveness of the latent space representation**. We test on steady state flow simulations in 3D, comparing against methods that use regular grid representations, and thus considerably larger latent space representations. (ii) **Scalability**. We test on transient flow simulations on large meshes. Specifically, we test the effectiveness of latent space rollouts, and assess how well UPTs generalize across different flow regime, and different domains, i.e., different number of mesh points and obstacles. (iii) **Lagrangian dynamics modeling**. Finally, we assess how well UPTs model the underlying field characteristics when applied to particle-based simulations. We outline the most important results in the following sections and provide implementation details and additional results in Appendix D. Most notably, UPT also compares favorably against baseline regular grid methods on regular grid datasets D.3.

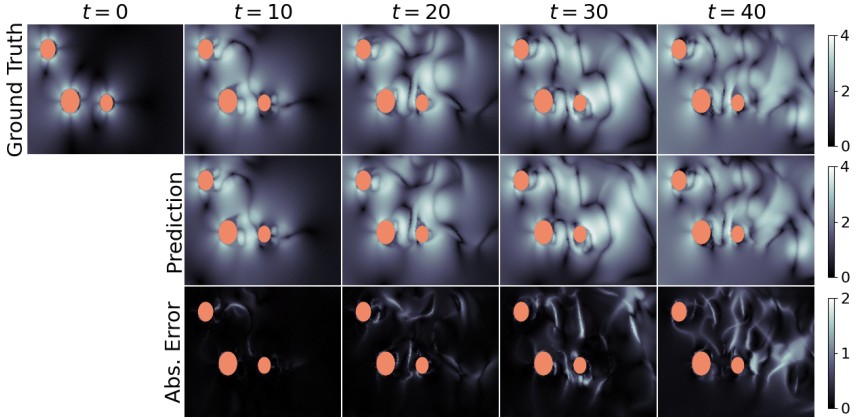

Figure 4: Example rollout trajectories of the UPT-68M model, visually demonstrating the efficacy of UPT physics modeling. The UPT model is trained across different obstacles, different flow regimes, and different mesh discretizations. Interestingly, the absolute error might suggest that UPT trajectories diverge, although physics are still simulated faithfully. This stems from subtle shifts in predictions throughout the rollout duration, likely attributed to the point-wise decoding of the latent field.

## 4.1 Steady state flows

For steady state prediction, we consider the ShapeNet-Car dataset generated by [105]. It consists of 889 car shapes from ShapeNet [20], where each car surface is represented by 3.6K mesh points in 3D space. We randomly create a train/test split containing 700/189 samples. We regress the pressure at each surface point with a mean-squared error (MSE) loss and sweep hyperparameters per model. Due to the small scale of this dataset, we train the largest possible model that is able to generalize the best. Training even larger models resulted in a performance decrease due to overfitting. We optimize the model size for all methods where the best mesh based models (GINO, UPT) contain around 300M parameters. The best regular grid based models (U-Net [86, 31], FNO [51]) are significantly smaller and range from 15M to 100M. Additional details are listed in Appendix D.4.

ShapeNet-Car is a small-scale dataset. Consequently, methods that map the mesh onto a regular grid can employ grids of *extremely* high resolution, such that the number of grid points is orders of magnitude higher than the number of mesh points. For example, a grid resolution of 64 points per spatial dimension results in 262K grid points, which is 73x the number of mesh points. As UPT is designed to operate directly on the mesh, we compare at different grid resolutions.

UPT is able to outperform models on smaller resolutions, for example UPT with only 64 latent tokens achieves a test MSE of 2.31 whereas GINO with resolution $48^3$ (110K tokens) achieves only 2.58 while taking significantly more runtime and memory. When additionally using feature engineering in the form of a signed distance function and $64^3$ grid resolution UPT achieves a competitive performance of 2.24 compared to 2.14 of GINO while remaining efficient. All test errors are multiplied by 100. UPT achieves a favorable cost-vs-performance tradeoff where the best GINO models requires 900 seconds per epoch whereas a only slightly worse (-0.17) UPT model takes only 4 seconds. We show a comprehensive table with all results in Appendix Tab. 4.

## 4.2 Transient flows

We test the scalability of UPTs on large-scale transient flow simulations. For this purpose, we self-generate 10K Navier-Stokes simulations within a pipe flow using the pisoFoam solver from OpenFOAM [110], which we split into 8K training, 1K validation and 1K test trajectories. For each simulation, between one and four objects (circles of variable size) are placed randomly within the pipe flow, and the uni-directional inflow velocity varies between 0.01 to 0.06 m/s. The simulation is carried out for 100s resulting in 100 timesteps that are used for training neural surrogates. Note that pisoFoam requires much smaller $\Delta t$ to remain stable (between 2K and 200K timesteps for 100s of simulation time). We use an adaptive meshing algorithm which results in 29K to 59K mesh points. Further dataset details are outlined in Appendix D.5.

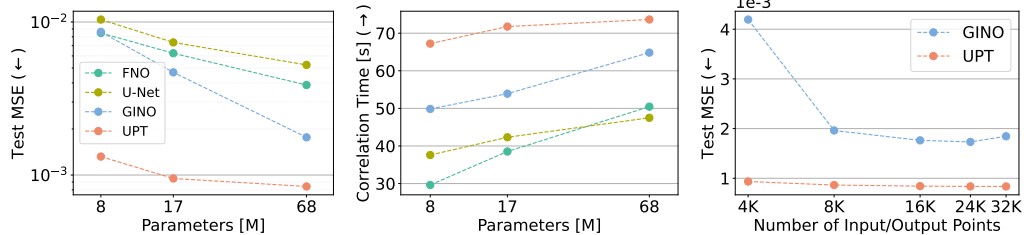

Figure 5: Left and middle: MSE and correlation time on the testset. UPTs outperform compared methods on all model scales by a large margin. Right: We study discretization convergence by varying the number of input/output points or the number of gridpoints/supernodes of models that were trained on inputs between 8K and 24K points, 8K target points. UPT demonstrates a stable performance across different number of input/outputs even though it has never seen that number of input/output points during training. We study discretization convergence of supernodes in App. D.5.2, where UPT also shows a steady improvement when more supernodes are used during inference. Additionally, we study smaller UPT models and training on less data in Appendix D.5.4 and D.5.5.

Model-wise, UPT uses the hierarchical encoder setup with all optional components depicted in Fig. 3. A message passing layer aggregates local information into $n_\text{s} = 2048$ randomly selected supernodes, a transformer processes the supernodes and a perceiver pools the supernodes into $n_\text{latent} = 512$ latent tokens. Approximator and decoder are unchanged. We compare UPT against GINO, U-Net and FNO. For U-Net and FNO, we interpolate the mesh onto a regular grid. We condition the models onto the current timestep and inflow velocity by modulating features within the model. We employ FiLM conditioning for U-Net [80], the "Spatial-Spectral" conditioning method introduced in [31] for FNO and GINO, and DiT for UPT [78]. Implementation details are provided in Appendix D.5.

We train all models for 100 epochs and evaluate test MSE as well as rollout performance for which we use the number of timesteps until the Pearson correlation of the rollout drops below 0.8 as evaluation metric [43]. We do not employ any techniques to stabilize rollouts [57]. The left side of Fig. 5, shows that UPTs outperform compared methods by a large margin. Training even larger models becomes increasingly expensive and is infeasible for our current computational budget. UPT-68M and GINO-68M training takes roughly 450 A100 hours. UPT-8M and UPT-17M take roughly 150 and 200 A100 hours, respectively. Figure 4 shows a rollout and Appendix D.5.1 presents additional ones. We also study out-of-distribution generalization (e.g. more obstacles) in Appendix D.5.6.

While one would ideally use lots of supernodes and query the latent space with all positions during training, increasing those quantities increases training costs and the performance gains saturate. Therefore, we only use 2048 supernodes and 16K randomly selected query positions during training. We investigate discretization convergence in the right part of Fig. 5 where we vary the number of input/output points and the number of supernodes. We use the 68M models *without* any retraining, i.e., we test models on "discretization convergence" as, during training, the mesh was discretized into 2048 supernodes and 16K query positions. UPT generalizes across a wide range of different number of input or output positions, with even slight performance increases when using more input points. Similarly, using more supernodes increases performance slightly. Additionally, we investigate training with more supernodes and/or more latent tokens in a reduced setting in Appendix D.5.3.

Finally, we evaluate training with inverse encoding and decoding techniques, see Fig. 3. We investigate the impact of the latent rollout by training our largest model – a 68M UPT. The latent rollout achieves on par results to autoregressively unrolling via the physics domain, but speeds up the inference significantly as shown in Tab. 1. However, in its current implementation the latent rollout requires a non-negligible overhead during training. We discuss this limitation in Appendix A.

### 4.3 Lagrangian fluid dynamics

Scaling particle-based methods such as discrete element methods or smoothed particle hydrodynamics to 10 million or more particles presents a significant challenge [117, 9], yet it also opens a distinctive opportunity for neural surrogates. Such systems are far beyond the scope of this work. We however present a framing of how to model such systems via UPTs such that the studied scaling properties

Table 1: Required time to simulate a full trajectory rollout. UPT and GINO are orders of magnitude faster than traditional finite volume solvers. The latent rollout is additionally more than 5x faster than an autoregressive rollout via the physics domain. Neural surrogate models are also faster on CPUs as traditional solvers require extremely small timescales to remain stable ($\Delta t \leqslant 0.05$ vs. $\Delta t = 1$).

| Model | Time on 16 CPUs | Time on 1 GPU | Speedup |
|---|---|---|---|
| pisoFoam | 120s | - | 1x |
| GINO-68M (autoreg.) | 48s | 1.2s | 100x |
| UPT-68M (autoreg.) | 46s | 2.0s | 60x |
| UPT-68M (latent) | 3s | **0.3s** | **400x** |

of UPTs could be exploited. In order to do so, we demonstrate how UPTs capture inherent field characteristics when applied to Lagrangian SPH simulations, as provided in LagrangeBench [101]. Here, GNNs, such as Graph Network-based Simulators (GNS) [87] and Steerable E(3) Equivariant Graph Neural Networks (SEGNNs) [14] are strong baselines, where predicted accelerations at the nodes are numerically integrated to model the time evolution of the particles. In contrast, UPTs learn underlying dynamics without dedicated particle-structures, and propagate dynamics forward without the guidance of numerical time integration schemes. An overview of the conceptual differences between GNS/SEGNN and UPTs is shown in Fig. 6.

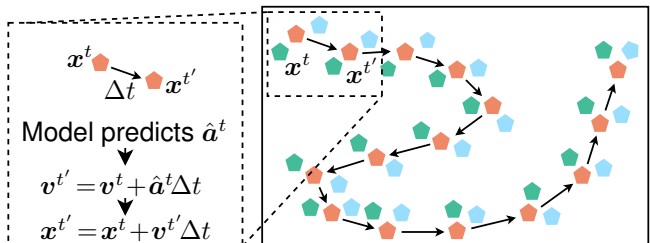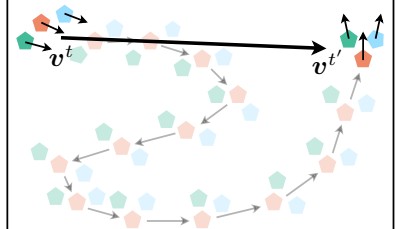

Figure 6: Conceptual difference between GNS/SEGNN on the left and UPT on the right side. GNS/SEGNN predicts the acceleration of a particle which is then integrated to calculate the next position. UPTs directly model the velocity field and allow for large timestep predictions.

We use the Taylor-Green vortex dataset in three dimensions (TGV3D). The TGV system was introduced as a test scenario for turbulence modeling [97]. It is an unsteady flow of a decaying vortex, displaying an exact closed form solution of the incompressible Navier–Stokes equations in Cartesian coordinates. We note that the TGV3D dataset models the same trajectories but does so by tracking particles using SPH to solve the equations. Formulating the TGV system as a UPT learning problem allows the same trajectories to be queried at different positions, enabling the recovery of the complete velocity field, whereas GNNs can only evaluate the velocities of the particles. Consequently, the evaluation against GNNs should be viewed as an illustration of the efficacy of UPTs in learning field characteristics, rather than a comprehensive GNN versus UPT comparison. More details and experiments on the 2D version of the Taylor-Green vortex dataset (TGV2D) are in Appendix D.6.

For UPT training, we input two consecutive velocities of the particles in the dataset at timesteps $t$ and $t - 1$, and the respective particle positions. We regress two consecutive velocities at a later timesteps $\{t' - 1, t'\} = \{t + \Delta T - 1, t + \Delta T\}$ with mean-squared error (MSE) objective. For all experiments we use $\Delta T = 10\Delta t$. We query the decoder to output velocities at target positions. UPTs encode the first two velocities of a trajectory, and autoregressively propagate dynamics forward in the latent space. We report the Euclidean norm of velocity differences across all $k$ particles. Figure 7 compares the rollout performance of GNS, SEGNN and UPT and shows the speedup of both methods compared to the SPH solver. The results demonstrate that UPTs effectively learn the underlying field dynamics.

## 5 Discussion

**Potential for extreme scale**. UPTs consist of mainly transformer [106] layers which allows application of the same scaling and parallelism principles as are commonly used for training extreme-scale language models. For example, the recent work Llama 3 [27] trained on up to 16K H100 GPUs.

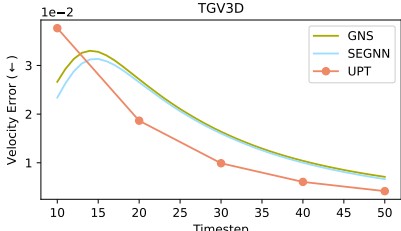 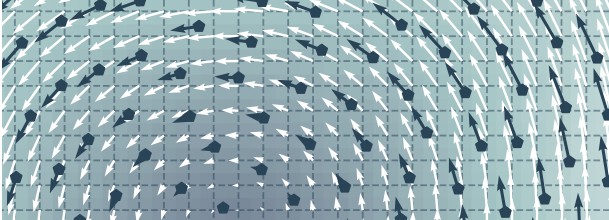

Figure 7: Left: Velocity error over all particles for different timesteps. UPTs effectively learn the underlying field dynamics, resulting in lower error as the trajectory evolves in time. Right: Visualization of the velocity field modeled by UPT (white) vs the ground truth particle velocities.

While we do not envision that we train UPT on such a massive scale in the foreseeable future, the used techniques can be readily applied to UPTs.

**Benefits of the latent rollout**. While the latent rollout does not provide a significant performance improvement, it is almost an order of magnitude faster. The UPT framework allows to trade-off training compute vs inference compute. If inference time is crucial for a given application, one can train UPT with the inverse encoding and decoding objectives, requiring more training compute but greatly speeding up inference. If inference time is not important, one can simply train UPT without the reconstruction objectives to reduce training costs.

Additionally, the latent rollout enables applicability to Lagrangian simulations. As UPT models the underlying field instead of tracking individual particle positions it does not have access to particle locations at inference time. Therefore, autoregressive rollouts are impossible since the encoder requires particle positions as input. When using the latent rollout, it is sufficient to know the initial particle positions as dynamics are propagated without any spatial positions. After propagating the latent space forward in time, the latent space can be queried at arbitrary positions to evaluate the underlying field at given positions. We show this by querying with regular grid positions in Figure 7.

## 6 Conclusion

We have introduced Universal Physics Transformers (UPTs) framework for efficiently scaling neural operators, demonstrating its applicability to a wide range of spatio-temporal problems. UPTs operate without grid- or particle-based latent structures, enabling flexibility across meshes and number of particles. The UPT training procedure separates responsibilities between components, allowing a forward propagation in time purely within the latent space. Finally, UPTs allow for queries of the latent space representation at any point in space-time.

## Acknowledgments

We would like to sincerely thank Artur P. Toshev and Gianluca Galletti for ongoing help with and in-depth discussions about LagrangeBench. Johannes Brandstetter acknowledges Max Welling and Paris Perdikaris for numerous stimulating discussions.

We acknowledge EuroHPC Joint Undertaking for awarding us access to Karolina at IT4Innovations, Czech Republic and Leonardo at CINECA, Italy.

The ELLIS Unit Linz, the LIT AI Lab, the Institute for Machine Learning, are supported by the Federal State Upper Austria. We thank the projects Medical Cognitive Computing Center (MC3), INCONTROL-RL (FFG-881064), PRIMAL (FFG-873979), S3AI (FFG-872172), DL for GranularFlow (FFG-871302), EPILEPSIA (FFG-892171), AIRI FG 9-N (FWF-36284, FWF-36235), AI4GreenHeatingGrids (FFG- 899943), INTEGRATE (FFG-892418), ELISE (H2020-ICT-2019-3 ID: 951847), Stars4Waters (HORIZON-CL6-2021-CLIMATE-01-01). We thank Audi.JKU Deep Learning Center, TGW LOGISTICS GROUP GMBH, Silicon Austria Labs (SAL), FILL Gesellschaft mbH, Anyline GmbH, Google, ZF Friedrichshafen AG, Robert Bosch GmbH, UCB Biopharma SRL, Merck Healthcare KGaA, Verbund AG, Software Competence Center Hagenberg GmbH, Borealis AG, TÜV Austria, Frauscher Sensonic, TRUMPF and the NVIDIA Corporation.

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

# A  Limitations and Future work

**Latent rollout.**   We show that the latent rollout can be enabled via a simple end-to-end training procedure, but ultimately think that it can be improved by more delicate training procedures such as a two stage procedure akin to diffusion models [85, 55]. As there are many possible avenues to apply or improve the latent rollout, it would exceed the scope of this paper and we therefore leave it for future work.

**Generalization beyond fluid dynamics.**   We show the generalization capabilities of UPTs across different simulation types and leave the application of UPTs to other domains for future work. We additionally show that UPTs are neural operators and since neural operators have been shown to generalize well across domains, this should also hold for UPTs.

**Large-scale Lagrangian simulations.**   A particularly intriguing direction is to apply UPTs to large-scale Lagrangian simulations. However, there do not exist readily available large-scale Lagrangian datasets. Therefore, we consider large-scale Lagrangian simulations beyond the scope of this paper as generating such a dataset requires extensive domain knowledge as well as engineering effort. Note that we show the favorable scaling properties of UPTs on the transient flow dataset, which contains up to 59K mesh points. In this setting, we train with a distributed setup of up to 32 A100 GPUs.

**Unifying Lagrangian and Eulerian simulations.**   UPTs can encode both Lagrangian and Eulerian simulations. It is therefore a natural future direction to exploit those different modalities. Especially, since particle- and grid-based simulations are used to describe different phenomena but model similar underlying dynamics. It is however to note that our method is designed primarily with the purpose of efficient scaling, and multi-modality training follows as side-concept thereof.

# B  Computational fluid dynamics

This appendix discusses the Navier-Stokes equations and selected numerical integration schemes, which are related to the experiments presented in this paper. This is not an complete introduction into the field, but rather encompasses the concepts which are important to follow the experiments discussed in the main paper.

First, we discuss the different operators of the Navier-Stokes equations, relate compressible and incompressible formulations, and discuss why Navier-Stokes equations are difficult to solve. Second, we discuss turbulence as one of the fundamental challenges of computational fluid dynamics (CFD), explicitly working out the difference for two dimensional and three dimensional turbulence modeling. Third, we introduce Reynolds-averaged Navier-Stokes equations (RANS) as an numerical approach for steady-state turbulence modeling, and the SIMPLE and PISO algorithms as steady-state and transient solvers, respectively. Finally, we discuss Lagrangian discretization schemes, focusing on smoothed particle hydrodynamics (SPH).

## B.1  Navier-Stokes equations

In our experiments, we mostly work with the incompressible Navier-Stokes equations [98]. In two spatial dimensions, the evolution equation of the velocity flow field $\boldsymbol{u} : [0, T] \times U \subset \mathbb{R}^3 \rightarrow \mathbb{R}^2$, $\boldsymbol{u}(t, x_1, x_2) = (u_1(t, x_1, x_2), u_2(t, x_1, x_2))$ with internal pressure $\boldsymbol{p}(x_1, x_2) = (p_1(x_1, x_2), p_2(x_1, x_2))$ and external force field $\boldsymbol{f}(x_1, x_2) = (f_1(x_1, x_2), f_2(x_1, x_2))$ is given via:

$$
\begin{aligned}
\frac{\partial u_1}{\partial t} &= - u_1 \frac{\partial u_1}{\partial x_1} - u_2 \frac{\partial u_1}{\partial x_2} + \mu \left( \frac{\partial^2 u_1}{\partial x_1^2} + \frac{\partial^2 u_1}{\partial x_2^2} \right) - \frac{\partial p_1}{\partial x_1} + f_1 \\
\frac{\partial u_2}{\partial t} &= - u_1 \frac{\partial u_2}{\partial x_1} - u_2 \frac{\partial u_2}{\partial x_2} + \mu \left( \frac{\partial^2 u_2}{\partial x_1^2} + \frac{\partial^2 u_2}{\partial x_2^2} \right) - \frac{\partial p_2}{\partial x_2} + f_2 \\
0 &= \frac{\partial u_1}{\partial x_1} + \frac{\partial u_2}{\partial x_2}
\end{aligned} \tag{3}
$$

This system is usually written in a shorter and more convenient form:

$$\frac{\partial \boldsymbol{u}}{\partial t} = -\boldsymbol{u} \cdot \nabla \boldsymbol{u} + \mu \nabla^2 \boldsymbol{u} - \nabla p + \boldsymbol{f}$$
$$0 = \nabla \cdot \boldsymbol{u}\,,$$

(4)

where

$$\boldsymbol{u} \cdot \nabla \boldsymbol{u} = \begin{pmatrix} u_1 & u_2 \end{pmatrix} \cdot \begin{pmatrix} \dfrac{\partial u_1}{\partial x_1} & \dfrac{\partial u_2}{\partial x_1} \\ \dfrac{\partial u_1}{\partial x_2} & \dfrac{\partial u_2}{\partial x_2} \end{pmatrix}$$

(5)

is called the convection, i.e., the rate of change of $\boldsymbol{u}$ along $\boldsymbol{u}$, and

$$\mu \nabla^2 \boldsymbol{u} = \mu \begin{pmatrix} \dfrac{\partial}{\partial x_1} & \dfrac{\partial}{\partial x_2} \end{pmatrix} \cdot \begin{pmatrix} \dfrac{\partial}{\partial x_1} \\ \dfrac{\partial}{\partial x_2} \end{pmatrix} \begin{pmatrix} u_1 \\ u_2 \end{pmatrix} = \mu \begin{pmatrix} \dfrac{\partial^2}{\partial x_1^2} + \dfrac{\partial^2}{\partial x_2^2} \end{pmatrix} \begin{pmatrix} u_1 \\ u_2 \end{pmatrix}$$

(6)

the viscosity, i.e., the diffusion or net movement of $\boldsymbol{u}$ with viscosity parameter $\mu$. The constraint

$$\begin{pmatrix} \dfrac{\partial}{\partial x_1} & \dfrac{\partial}{\partial x_2} \end{pmatrix} \cdot \begin{pmatrix} u_1 \\ u_2 \end{pmatrix} = \frac{\partial u_1}{\partial x_1} + \frac{\partial u_2}{\partial x_2} = 0$$

(7)

yields mass conservation of the Navier-Stokes equations.

The incompressible version of the NS equations above are commonly used in the study of water flow, low-speed air flow, and other scenarios where density changes are negligible. Its compressible counterpart additionally accounts for variations in fluid density and temperature which is necessary to accurately model gases at high speeds (starting in the magnitude of speed of sound) or scenarios with significant changes in temperature. These additional considerations result in a more complex momentum and continuity equation.

**Why are the Navier-Stokes equations difficult to solve?** One major difficulty is that no equation explicitly models the unknown pressure field. Instead, the conservation of mass $\nabla \cdot \boldsymbol{u} = 0$ is an implicit constraint on the velocity fields. Additionally, the nonlinear nature of the NS equations makes them notoriously harder to solve than parabolic or hyperbolic PDEs such as the heat or wave equation, respectively. Also, the occurrence of turbulence as discussed in the next subsection which is due to the fact that small and large scales are coupled such that small errors, can have a large effect on the computed solution.

**Computational fluid dynamics** (CFD) uses numerical schemes to discretize and solve fluid flows. CFD simulates the free-stream flow of the fluid, and the interaction of the fluid (liquids and gases) with surfaces defined by boundary conditions. As such, CFD comprises many challenging phenomena, such as interactions with boundaries, mixing of different fluids, transonic, i.e, coincident emergence of subsonic and supersonic airflow, or turbulent flows.

## B.2 Turbulence

Turbulence [82, 64] is one of the key aspects of CFD and refers to chaotic and irregular motion of fluid flows, such as air or water. It is characterized by unpredictable changes in velocity, pressure, and density within the fluid. Turbulent flow is distinguished from laminar flow, which is smooth and orderly. There are several factors that can contribute to the onset of turbulence, including high flow velocities, irregularities in the shape of surfaces over which the fluid flows, and changes in the fluid's viscosity. Turbulence plays a significant role in many natural phenomena and engineering applications, influencing processes such as mixing, heat transfer, and the dispersion of particles in fluids. Understanding and predicting turbulence is crucial in fields like fluid dynamics, aerodynamics, and meteorology. Turbulent flows occur at high Reynolds numbers

$$\mathrm{Re} = \frac{\rho \nu L}{\mu}\,,$$

(8)

where $\rho$ is the density of the fluid, $\nu$ is the velocity of the fluid, $L$ is the linear dimension, e.g., the width of a pipe, and $\mu$ is the viscosity parameter. Turbulent flows are dominated by inertial forces, which tend to produce chaotic eddies, vortices and other flow instabilities.

**Turbulence in 3D vs turbulence in 2D**. For high Reynolds numbers, i.e., in the turbulent regime, solutions to incompressible Navier-Stokes equations differ in three dimensions compared to two dimensions by a phenomenon called *energy cascade*. Energy cascade orchestrates the transfer of kinetic energy from large-scale vortices to progressively smaller scales until it is eventually converted into thermal energy. In three dimensions, this energy cascade is responsible for the continuous formation of vortex structures at various scales, and thus for the emergence of high-frequency features. In contrast to three dimensions, the energy cascade is inverted in two dimensions, i.e., energy is transported from smaller to larger scales, resulting in more homogeneous, long-lived structures. Mathematically the difference between three dimensional and two dimensional turbulence modeling can be best seen by rewriting the Navier-Stokes equations in vorticity formulation, where the vorticity $\boldsymbol{\omega}$ is the curl of the flow field, i.e., $\boldsymbol{\omega} = \nabla \times \boldsymbol{u}$, with $\times$ here denoting the cross product. This derivation can be found in many standard texts on Navier-Stokes equation , e.g. Tsai [104, Section 1.4.], and for the reader's convenience we briefly repeat the derivation here. We start using Eq. (4), setting $\boldsymbol{f} = 0$ for simplicity:

$$\frac{\partial \boldsymbol{u}}{\partial t} = -\boldsymbol{u} \cdot \nabla \boldsymbol{u} + \mu \nabla^2 \boldsymbol{u} - \nabla p \tag{9}$$

$$= -\nabla \left( \frac{1}{2} \boldsymbol{u} \cdot \boldsymbol{u} \right) + \boldsymbol{u} \times \nabla \times \boldsymbol{u} + \mu \nabla^2 \boldsymbol{u} - \nabla p , \tag{10}$$

where we applied the dot product rule for derivatives of vector fields:

$$\nabla \left( \frac{1}{2} \boldsymbol{u} \cdot \boldsymbol{u} \right) = \boldsymbol{u} \cdot \nabla \boldsymbol{u} + \boldsymbol{u} \times \nabla \times \boldsymbol{u} . \tag{11}$$

Next, we take the curl of the right and the left hand side:

$$\frac{\partial}{\partial t} (\nabla \times \boldsymbol{u}) = -\frac{1}{2} \underbrace{\nabla \times \nabla (\boldsymbol{u} \cdot \boldsymbol{u})}_{=0} + \nabla \times \boldsymbol{u} \times \nabla \times \boldsymbol{u} + \mu \nabla^2 (\nabla \times \boldsymbol{u}) - \underbrace{\nabla \times \nabla p}_{=0} \tag{12}$$

$$\frac{\partial \boldsymbol{\omega}}{\partial t} = \nabla \times (\boldsymbol{u} \times \boldsymbol{\omega}) + \mu \nabla^2 \boldsymbol{\omega} \tag{13}$$

using the property that the curl of a gradient is zero. Lastly, using that the divergence of the curl is again zero (i.e. $\nabla \cdot \boldsymbol{\omega} = 0$), via the identity $\nabla \times (\boldsymbol{u} \times \boldsymbol{\omega}) = (\boldsymbol{\omega} \cdot \nabla) \boldsymbol{u} - (\boldsymbol{u} \cdot \nabla) \boldsymbol{\omega} + \boldsymbol{u} \underbrace{\nabla \cdot \boldsymbol{\omega}}_{=0} - \boldsymbol{\omega} \underbrace{\nabla \cdot \boldsymbol{u}}_{=0}$,

we obtain:

$$\frac{\partial \boldsymbol{\omega}}{\partial t} + (\boldsymbol{u} \cdot \nabla) \boldsymbol{\omega} = (\boldsymbol{\omega} \cdot \nabla) \boldsymbol{u} + \mu \nabla^2 \boldsymbol{\omega} \tag{14}$$

$$\frac{D \boldsymbol{\omega}}{Dt} = (\boldsymbol{\omega} \cdot \nabla) \boldsymbol{u} + \mu \nabla^2 \boldsymbol{\omega} , \tag{15}$$

with the material derivative $\frac{D \boldsymbol{\omega}}{Dt} = \frac{\partial \boldsymbol{\omega}}{\partial t} + (\boldsymbol{u} \cdot \nabla) \boldsymbol{\omega}$.

For three dimensional flows the material derivative of each component $\omega_i$ can be written as

$$\frac{D \omega_i}{Dt} = \sum_{j=1}^{3} \left( \underbrace{\omega_j \frac{\partial u_i}{\partial x_j}}_{\text{vortex turning and stretching}} + \underbrace{\mu \frac{\partial^2 \omega_i}{\partial x_j \partial x_j}}_{\text{diffusion}} \right) , \tag{16}$$

for example picking $i = 2$:

$$\frac{D \omega_2}{Dt} = \underbrace{\omega_1 \frac{\partial u_2}{\partial x_1}}_{\text{vortex turning}} + \underbrace{\omega_2 \frac{\partial u_2}{\partial x_2}}_{\text{vortex stretching}} + \underbrace{\omega_3 \frac{\partial u_2}{\partial x_3}}_{\text{vortex turning}} + \underbrace{\mu \left( \frac{\partial^2 \omega_2}{\partial x_1 \partial x_1} + \frac{\partial^2 \omega_2}{\partial x_2 \partial x_2} + \frac{\partial^2 \omega_2}{\partial x_3 \partial x_3} \right)}_{\text{diffusion}} . \tag{17}$$

For two-dimensional flows, as represented by

$$\boldsymbol{u} = \begin{pmatrix} u_1 \\ u_2 \\ 0 \end{pmatrix} \text{ and } \boldsymbol{\omega} = \nabla \times \boldsymbol{u} = \begin{pmatrix} 0 \\ 0 \\ \frac{\partial u_2}{\partial x_1} - \frac{\partial u_1}{\partial x_2} \end{pmatrix} , \tag{18}$$

the terms for vortex turning and vortex stretching vanish, as is evident in the material derivative of $\omega_3$, given by

$$\frac{D\omega_3}{Dt} = \underbrace{\omega_1 \frac{\partial u_3}{\partial x_1}}_{=0} + \underbrace{\omega_2 \frac{\partial u_3}{\partial x_2}}_{=0} + \underbrace{\omega_3 \frac{\partial u_3}{\partial x_3}}_{=0} + \underbrace{\mu \left( \frac{\partial^2 \omega_3}{\partial x_1 \partial x_1} + \frac{\partial^2 \omega_3}{\partial x_2 \partial x_2} \right)}_{\text{diffusion}} + \underbrace{\mu \frac{\partial^2 \omega_3}{\partial x_3 \partial x_3}}_{=0} . \tag{19}$$

Consequently, the length and the angle of a vortex do not change in two dimensions, resulting in homogeneous, long-lived structures.

There's been several approaches to model turbulence with the help of Deep Learning. Predominantly models for two dimensional scenarios have been suggested [81, 51, 15, 43, 44]. Due to the higher complexity (as discussed above) and memory- and compute costs comparably less work was done in the 3D case [92, 56].

## B.3 Numerical modeling based on Eulerian discretization schemes

Various approaches are available for numerically addressing turbulence, with different schemes suited to different levels of computational intensity. Among these, Direct Numerical Simulation (DNS) [68] stands out as the most resource-intensive method, involving the direct solution of the unsteady Navier-Stokes equations.

DNS is renowned for its capability to resolve even the minutest eddies and time scales present in turbulent flows. While DNS does not necessitate additional closure equations, a significant drawback is its high computational demand. Achieving accurate solutions requires the utilization of very fine grids and extremely small time steps. This is based on Kolmoqorov [46] and Kolmogorov [45] which give the minimum spacial and temporal scales that need to be resolved for accurately simulating turbulence. The discretization-scales for both time and space needed for 3D-problems become very small and simulations computationally extremely expensive.

This computational intensity of DNS resulted in the development of turbulence modeling, with Large Eddy Simulations (LES) [91] and Reynolds-averaged Navier-Stokes equations (RANS) [84] being two prominent examples. LES aim to reduce computational cost by neglecting the computationally expensive smallest length scales in turbulent flows. This is achieved through a low-pass filtering of the Navier–Stokes equations, effectively removing fine-scale details via time- and spatial-averaging. In contrast, the Reynolds-averaged Navier–Stokes equations (RANS equations) are based on time-averaging. The foundational concept is Reynolds decomposition, attributed to Osborne Reynolds, where an instantaneous quantity is decomposed into its time-averaged and fluctuating components.

Turbulent scenarios where the fluid conditions change over time are an example of transient flows. These are typically harder to model/solve than steady state flows where the fluid properties exhibit only negligible changes over time.

In the following, we discuss Reynolds-averaged Navier-Stokes equations (RANS) as a numercial approach for steady-state turbulence modeling, and the SIMPLE and PISO algorithms as steady-state and transient numerical solvers, respectively.

### B.3.1 Reynolds-averaged Navier-Stokes equations

Reynolds-averaged Navier-Stokes equations (RANS) are used to model time-averaged fluid properties such as velocity or pressure that result in a steady state which does not change over time. Writing (4) in Einstein notation we get

$$\frac{\partial u_i}{\partial t} = -u_j \frac{\partial u_i}{\partial x_j} + \mu \frac{\partial^2 u_i}{\partial x_j \partial x_j} - \frac{\partial p}{\partial x_i} + f_i$$

$$0 = \frac{\partial u_i}{\partial x_i} . \tag{20}$$

Taking the Reynolds decomposition [84] $g(t, x_1, x_2) := \bar{g}(x_1, x_2) + g'(t, x_1, x_2)$ with $\bar{g}(x_1, x_2) :=$ $\lim_{T \to \infty} 1/T \int_0^T g(x_1, x_2, t)dt$ being the time-average on $[0, T]$ of a scalar valued function $g$, and splitting each term of both equations into it's time-averaged and fluctuating part we get

$$\frac{\partial(\bar{u}_i + u_i')}{\partial t} = -(\bar{u}_j + u_j')\frac{\partial(\bar{u}_i + u_i')}{\partial x_j} + \mu\frac{\partial^2(\bar{u}_i + u_i')}{\partial x_j \partial x_j} - \frac{\partial(\bar{p} + p')}{\partial x_i} + (\bar{f}_i + f_i')$$

$$0 = \frac{\partial(\bar{u}_i + u_i')}{\partial x_i}.$$

Time-averaging these equations together with the property that the time-average of the fluctuating parts equals zero results in

$$\bar{u}_j\frac{\partial \bar{u}_i}{\partial x_j} + \overline{u_j'\frac{\partial u_i'}{\partial x_j}} = \mu\frac{\partial^2 \bar{u}_i}{\partial x_j \partial x_j} - \frac{\partial \bar{p}}{\partial x_i} + \bar{f}_i \tag{21}$$

$$0 = \frac{\partial \bar{u}_i}{\partial x_i}.$$

Using the the mass conserving equation (20) the momentum equation (21) can be rewritten as

$$\bar{u}_j\frac{\partial \bar{u}_i}{\partial x_j} = \bar{f}_i + \frac{\partial}{\partial x_j}\left[\mu\left(\frac{\partial \bar{u}_i}{\partial x_j} + \frac{\partial \bar{u}_j}{\partial x_i}\right) - \bar{p}\delta_{ij} - \overline{u_i'u_j'}\right]$$

where $-\overline{u_i'u_j'}$ is called the Reynolds stress which needs further modeling to solve the above equations. More specifically, this is referred to as the Closure Problem which led to many turbulence models such as $k$-$\epsilon$ [32] or $k$-$\omega$ [112]. Analogously the 3D RANS equations can be derived. This turbulence model consists of simplified equations that predict the statistical evolution of turbulent flows. Due to the Reynolds stress there still remain velocity fluctuations in the RANS equations. To get equations that contain only time-averaged quantities the RANS equations need to be closed by modeling the Reynolds stress as a function of the mean flow such that any reference to the fluctuating parts is removed. The first such approach led to the eddy viscosity model [11] for 3-d incompressible Navier Stokes:

$$-\overline{u_i'u_j'} = \nu_t\left(\frac{\partial \bar{u}_i}{\partial x_j} + \frac{\partial \bar{u}_j}{\partial x_i}\right) - \frac{2}{3}k\delta_{ij}$$

with the turbulence eddy viscosity $\nu_t > 0$ and the turbulence kinetic energy $k = 1/2\ \overline{u_i'u_i'}$ based on Boussinesq's hypothesis that turbulent shear stresses act in the same direction as shear stresses due to the averaged flow. The $k$-$\epsilon$ model employs Boussinesq's hypothesis by using comparably low-cost computations for the eddy viscosity by means of two additional transport equations for turbulence kinetic energy $k$ and dissipation $\epsilon$

$$\frac{\partial(\rho k)}{\partial t} + \frac{\partial(\rho k u_i)}{\partial x_i} = \frac{\partial}{\partial x_j}\left[\frac{\nu_t}{\sigma_k}\frac{\partial k}{\partial x_j}\right] + 2\nu_t E_{ij}E_{ij} - \rho\epsilon$$

$$\frac{\partial(\rho\epsilon)}{\partial t} + \frac{\partial(\rho\epsilon u_i)}{\partial x_i} = \frac{\partial}{\partial x_j}\left[\frac{\nu_t}{\sigma_\epsilon}\frac{\partial \epsilon}{\partial x_j}\right] + C_{1\epsilon}\frac{\epsilon}{k}2\nu_t E_{ij}E_{ij} - C_{2\epsilon}\rho\frac{\epsilon^2}{k}$$

with the rate of deformation $E_{ij}$, eddy viscosity $\nu_t = \rho C_\nu k^2/\epsilon$, and adjustable constants $C_\nu, C_{1\epsilon}, C_{2\epsilon}, \sigma_\epsilon, \sigma_k$. In our experiment section 4.1 solutions of simulations based on the RANS $k$-$\epsilon$ turbulence model are used as ground truth where the quantity of interest is the pressure field given the shape of a vehicle.

### B.3.2   SIMPLE and PISO algorithm

Next, we introduce two popular algorithms that try to solve Eq. (4) numerically, where the main idea is to couple pressure and velocity computations. Specifically, we will discuss the SIMPLE (Semi-Implicit Method for Pressure-Linked Equations) [77] and PISO (Pressure Implicit with Splitting of Operators) [38] algorithm, which are implemented in OpenFOAM as well. The essence of these algorithms are the following four main steps:

1. In a first step, the momentum equation is discretized by using suitable finite volume discretizations of the involved derivatives. This leads to a linear system for $\boldsymbol{u}$ for a given pressure gradient $\nabla\boldsymbol{p}$:

$$\mathcal{M}\boldsymbol{u} = -\nabla\boldsymbol{p}. \tag{22}$$

   However, this computed velocity field $\boldsymbol{u}$ does not yet satisfy the continuity equations $\nabla \cdot \boldsymbol{u} = 0$.

2. In a next step, let us denote by $\mathcal{A}$ the diagonal part of $\mathcal{M}$ and introduce $\mathcal{H}$, so that

$$\mathcal{H} = \mathcal{A}\boldsymbol{u} - \mathcal{M}\boldsymbol{u}. \tag{23}$$

   Taking into account $\mathcal{M}\boldsymbol{u} = -\nabla\boldsymbol{u}$ allows to easily rearrange Eq. (23) for $\boldsymbol{u}$, since $\mathcal{A}$ is easily invertible. This gives us:

$$\boldsymbol{u} = \mathcal{A}^{-1}\mathcal{H} - \mathcal{A}^{-1}\nabla\boldsymbol{p}. \tag{24}$$

3. Now we substitute Eq. (24) into the continuity equation yielding

$$\nabla \cdot (\mathcal{A}^{-1}\nabla\boldsymbol{p}) = \nabla \cdot (\mathcal{A}^{-1}\mathcal{H}), \tag{25}$$

   which is a Poisson equation for the pressure $\boldsymbol{p}$ that can be solved again numerically.

4. This pressure field can now be used to correct the velocity field by again applying Eq. (24). This is the pressure-corrector stage and this $\boldsymbol{u}$ now satisfies the continuity equation. Now, however, the pressure equation is no longer valid, since $\mathcal{H}$ depends on $\boldsymbol{u}$ as well. A way out here is iterating these procedures, which is the main idea of the SIMPLE and PISO algorithm.

The SIMPLE algorithm just iterates steps 1-4 several times , which is then called an *outer corrector loop*. In contrast, the PISO algorithm solves the momentum predictor $\mathcal{M}\boldsymbol{u} = -\nabla\boldsymbol{p}$ (i.e. step 1) only once and then iterates steps 2-4, which then result in an *inner loop*. In both algorithms, in case the meshes are non orthogonal, step 3, i.e. solving the Poisson equation, can also be repeated several times before moving to step 4. For stability reasons, the SIMPLE algorithm is preferred for stationary equations (since it implicitly promotes under-relaxation), whereas in case of time-dependent flows, one usually considers the PISO algorithm (since in practice, for each timestep several thousands of iterations are usually required until convergence, which is computationally very expensive).

## B.4 Lagrangian discretization schemes

In contrast to such grid- and mesh-based representations, in Lagrangian schemes, the discretization is carried out using finitely many material points, often referred to as particles, which move with the local deformation of the continuum. Roughly speaking, there are three families of Lagrangian schemes: discrete element methods [24], material point methods [94, 12], and smoothed particle hydrodynamics (SPH) [29, 62, 69, 70].

### B.4.1 Smoothed particle hydrodynamics

The core idea behind SPH is to divide a fluid into a set of discrete moving elements referred to as particles. These particles interact through using truncated radial interpolation kernel functions with characteristic radius known as the smoothing length. The truncation is justified by the assumption of locality of interactions between particles which allows to approximate the properties of the fluid at any arbitrary location. For particle $i$ its position $\boldsymbol{x}_i$ is given by its velocity $u_i$ such that $\frac{\partial \boldsymbol{x}_i}{\partial t} = \boldsymbol{u}_i$. The modeling of particle interaction by kernel functions implies that physical properties of any particle can be obtained by aggregating the relevant properties of all particles within the range of the kernel. For example, the density $\rho_i$ of a particle can be expressed as $\rho(\boldsymbol{x}_i) = \sum_j \rho_j V_j W(||\boldsymbol{x}_i - \boldsymbol{x}_j||, h)$ with $W$ being a kernel, $h$ its smoothing length, and $V_j$ the volumes of the respective particles. Rewriting the weakly-compressible Navier-Stokes equations with quantities as in (4) and additional Reynolds number Re and density $\rho$

$$\frac{\partial \boldsymbol{u}}{\partial t} = \frac{1}{\text{Re}}\nabla^2\boldsymbol{u} - \frac{1}{\rho}\nabla p + \frac{1}{\rho}\boldsymbol{f}$$

$$\frac{\partial \rho}{\partial t} = -\rho(\nabla \cdot \boldsymbol{u})$$

in terms of these kernel interpolations leads to a system of ordinary differential equations (ODEs) for the particle accelerations [72] where the respective velocities and positions can be computed by integration. One of the advantages of SPH compared to Eulerian discretization techniques is that SPH can handle large topological changes as no connectivity constraints between particles are required, and advection is treated exactly [100, 71]. The lack of a mesh significantly simplifies the model implementation and opens up more possibilities for parallelization compared to Eulerian schemes [35, 23]. However, for accurately resolving particle dynamics on small scales a large number of particles is needed to achieve a resolution comparable to grid-based methods specifically when the metric of interest is not directly related to density which makes SPH more expensive [22]. Also, setting boundary conditions such as inlets, outlets, and walls is more difficult than for grid-based methods.

A seemingly great fit for particle-based dynamics are graph neural networks (GNNs) [88, 42] with graph-based latent space representations. In many cases, predicted accelerations at the nodes are numerically integrated to model the time evolution of the many-particle systems [87, 65, 100, 102].

## C  Justification that UPTs are universal neural operators

We provide a brief sketch of how universality can be established for our architecture. For transformer-based neural operators, universal approximation has been recently demonstrated in [17], Section 5. The arguments in this work are heavily based on [49], which establishes that nonlinearity and nonlocality are crucial for universality. By demonstrating that the attention mechanism can, under appropriate weight choices, function as an averaging operator, the results from [49] are directly applicable. For detailed proof, refer to Theorem 22 in [17]. Our algorithm fits within this framework as well: we employ nonlinear, attention-based encoders and decoders (as allowed by the results of [49], Section 2) and utilize attention layers in the latent space.

## D  Experimental details and extended results

### D.1  General

All experiments are conducted mostly on A100 GPUs. For large-scale experiments we use two research clusters equipped with either 8xA100-40GB nodes or 4xA100-64GB nodes. For smaller experiments and evaluations, we use a mix of internal servers equipped with varying numbers of A100-40GB or A100-80GB cards.

We estimate the total number of GPU-hours (mostly A100-hours) used for this project to be 45K. This includes everything from model architecture design, exploratory training runs, investigating training instabilities to training/evaluating baseline models and UPTs.

All experiments linearly scale the learning rate with the batchsize, exclude normalization and bias parameters from the weight decay, follow a linear warmup $\rightarrow$ cosine decay learning rate schedule [58] and use the AdamW [41, 59] optimizer. Transformer blocks follow a standard pre-norm architecture as used in ViT [26].

### D.2  Feature modulation

We apply feature modulation to condition models to time/velocity. We use different forms of feature modulation in accordance with existing works depending on the model type. FiLM [80] is used for U-Net, the "Spatial-Spectral" conditioning of [31] is used for FNO based models and DiT [78] is used for transformer blocks. For perceiver blocks, we extend the modulation of DiT to seperately modulate queries from keys/values.

### D.3  Experiments on regular grid datasets

We use two datasets from [31], the "Navier-Stokes 2D" and "ShallowWater-2D" to compare against baseline transformer architectures. Baseline results, data preprocessing and benchmark setups are taken from DPOT [34] and CViT [108].

As these datasets contain only regular grid data, we make the following modifications to UPT: (i) we replace the supernode pooling with a patch embedding (ii) we remove the perceiver pooling in the encoder (iii) we decode patchwise. These modifications were done as (i) patch embedding is more efficient than supernode pooling for regular grid data (ii) these datasets are small-scale datasets, so compressing the latent space to make the model more efficient is not necessary (iii) as the decoding positions are always the same, there is no reason to decode position-wise as the missig variability of positions in the dataset makes superresolution impossible and patch-wise decoding is more efficient than position-wise decoding. This setup is conceptually similar to CViT [108].

We use a peak learning rate of 1e-4 with 10% warmup and a cosine decay schedule afterwards, patchsize 8, batchsize 256 and train for 1000 epochs. We use hidden dimension 96 and 12 blocks for 2M parameter models and hidden dimension 256 with 14 blocks for 13M parameter models to best match model sizes of previous models. Additionally, we mask out between 0% and 5% of the input patches via an attention mask in the transformer and perceiver blocks. Similar performances could be achieved with less epochs, but our flexible masking strategy allows long training without overfitting.

### D.3.1 NavierStokes-2D dataset

We run comparisons against different Transformer baselines on a regular gridded Navier-Stokes equations dataset [31] in Table 2 (baseline results and evaluation protocol taken from DPOT [34]). UPT outperforms all compared methods, some of which are specifically designed for regularly gridded data.

As baseline transformers often train small models, we compare on a small scale, where UPT significantly outperforms other models. We also compare on larger scales, where UPT again outperforms competitors.

Table 2: Comparison on a regular grid Navier-Stokes dataset [31]. **(a)** UPT outperforms competitor methods that train only small models by a large margin. **(b)** UPT also performs well on larger model sizes, outperforming competitors even if they train much larger models or pre-train (PT) on more data followed by fine-tuning (FT) on the Navier-Stokes dataset.

| (a) **Comparison on small model sizes** | | | (b) **Comparison on larger model sizes** | | |
|---|---|---|---|---|---|
| Model | # Params | Rel. L2 Error | Model | # Params | Rel. L2 Error |
| FNO [51] | 0.5M | 9.12% | DPOT-Ti [34] | 7M | 12.50% |
| FFNO [103] | 1.3M | 8.39% | DPOT-S [34] | 30M | 9.91% |
| GK-T [18] | 1.6M | 9.52% | DPOT-L (PT) | 500M | 7.98% |
| GNOT [33] | 1.8M | 17.20% | DPOT-L (FT) | 500M | 2.78% |
| OFormer [53] | 1.9M | 13.50% | DPOT-H (PT) | 1.03B | 3.79% |
| UPT-T | 1.8M | **5.08%** | CViT-S [108] | 13M | 3.75% |
| | | | CViT-B [108] | 30M | 3.18% |
| | | | UPT-S | 13M | 3.12% |
| | | | UPT-B | 30M | **2.69%** |

### D.3.2 ShallowWater-2D dataset

We run comparisons against UNet, FNO, Dilated ResNet variants on the regular gridded ShallowWater-2D climate modeling dataset [31] in Table 3 (baseline results and evaluation protocol taken from CViT [108]). UPT outperforms all compared methods, which are specifically designed for regularly gridded data.

### D.4 ShapeNet-Car

**Dataset**. We test on the dataset generated by [105], which consists 889 car shapes from ShapeNet [20] where side mirrors, spoilers and tires were removed. We randomly split the 889 simulations into 700 train and 189 test samples. Each sample consists of 3682 mesh points, including a small amount of points that are not part of the car mesh. We filter all points that do not belong to the car mesh, resulting in 3586 points per sample. [105] simulated 10 seconds of air flow and averaged the results

Table 3: Comparison on the regular gridded small-scale ShallowWater-2D climate modeling dataset. UPT outperforms models that are specifically designed for regular grid data.

| Model | # Params | Rel. L2 Error |
|---|---|---|
| DilResNet [93] | 4.2M | 13.20% |
| U-Net [86, 31] | 148M | 5.68% |
| FNO [51] | 268M | 3.97% |
| CViT-S [108] | 13M | 4.47% |
| UPT-S | 13M | **3.96%** |

the last 4 seconds. The inflow velocity is fixed at $20$ m/s with an estimated Reynolds Number of $\text{Re} = 5 \times 10^6$. For each point, we predict the associated pressure value. As there is no notion of time and the inflow velocity is constant, no feature modulation is necessary. We use the transformer positional encoding [106], and, therefore rescale all positions to the range $[0, 200]$. We normalize the pressure values to have zero mean and unit variance.

**Baselines**. As baselines we consider FNO, U-Net and GINO. For FNO, we use a hidden dimension of 64 and 16/24/24 Fourier modes per dimension, and train with a learning rate of 1e-3/1e-3/5e-4 for grid resolutions $32^3/48^3/64^3$, respectively. We also tried using 32 Fourier modes for a grid resolution of $64^3$, which performed worse than using 24 modes.

The U-Net baseline follows the architecture used as baseline in GINO which consists of 4 downsampling and 4 upsampling blocks where each block consists of two Group-Norm [115]$\rightarrow$Conv3d$\rightarrow$ReLU [74] subblocks. The initial hidden dimension is set to 64 then doubled in each downsampling block and halved in each upsampling block. Similar to FNO, we considered a higher initial hidden dimension for grid resolution $64^3$ which performed worse.

For GINO, we create a positional embedding of mesh- and grid-positions which are then used as input to a 3 layer MLP with GELU [37] non-linearities to create messages. Messages from mesh-positions within a radius of each grid-position are accumulated which serves as input to the FNO part of GINO. The FNO uses 64 hidden dimensions with 16/24/32 modes per dimension for grid resolutions $32^3/48^3/64^3$, respectively. The GINO decoder encodes the query positions and again uses a 3 layer MLP with GELU non-linearities to create messages. Each query position aggregates the messages from grid-points within a radius. The radius for message aggregation is set to $10$. When using SDF features, the SDF features are encoded via a 2 layer MLP to the same dimension as the hidden dimension, and added onto the positional encoding of each grid point.

**Hyperparameters**. We train for 1000 epochs with 50 warmup epochs, a batchsize of 32. We tune the learning rate and model size for each model and report the best test loss.

**UPT architecture**. Due to a relatively low number of mesh points (3586), we use only a single perceiver block with 64 learnable query tokens as encoder, followed by the standard UPT architecture (transformer blocks as approximator and another perceiver block as decoder).

When additionally using SDF features as input, the SDF features are encoded by a shallow ConvNeXt V2 [113] that processes the SDF features into 512 (8x8x8) tokens, which are concatenated to the perceiver tokens. To distinguish between the two token types, a learnable vector per type is added to each of the tokens.

As UPT operates on the mesh directly, we can augment the data by randomly dropping mesh points of the input. Therefore we randomly sample between 80% and 100% of the mesh points during training and evaluate with all mesh points when training without SDF features. The decoder still predicts the pressure for all mesh points by querying the latent space with each mesh position. We also tried randomly dropping mesh points for GINO where it degraded performance.

**Interpolated models**. We consider U-Net and FNO as baseline models. These models only take SDF features as input since both models are bound to a regular grid representation, which prevents using the mesh points directly. The output feature maps are linearly interpolated to each mesh point from which a shallow MLP predicts the pressure at each mesh point. In the setting where no SDF features are used, we assign each mesh point a constant value of 1 and linearly interpolate onto a grid.

**Extended results**. Table 4 gives a detailed overview of all results from ShapeNet-Car with varying resolutions and comparison to baseline models.

Table 4: Normalized test MSE for ShapeNet-Car pressure prediction. Memory is the amount of memory required to train on a single sample. UPTs can model the dynamics with a fraction of latent tokens compared to other models. SDF additionally uses the signed distance function from each gridpoint to the geometry as input features. To include SDF features into UPT, we encode the SDF features with resolution $32^3$, $48^3$ or $64^3$ into $8^3$ tokens using a shallow ConvNeXt V2 [113] and concatenate these tokens to the tokens coming from the UPT encoder. To balance the number of SDF tokens with the number of latent tokens, we increase the number of latent tokens to 1024 in the settings where we use SDF features for UPT. Runtime is measured on an A100 GPU.

| Model | SDF | #Tokens | MSE [1e-2] | Mem. [GB] | Runtime per Epoch [s] |
|---|---|---|---|---|---|
| U-Net | ✗ | $64^3$ | 6.13 | 1.3 | 86 |
| FNO | ✗ | $64^3$ | 4.04 | 3.8 | 148 |
| GINO | ✗ | $64^3$ | 2.34 | 19.8 | 900 |
| UPT (ours) | ✗ | 64 | **2.31** | **0.6** | **4** |
| U-Net | ✓ | $32^3$ | 3.66 | 0.2 | 11 |
| FNO | ✓ | $32^3$ | 3.31 | 0.5 | 17 |
| GINO | ✓ | $32^3$ | 2.90 | 2.1 | 103 |
| UPT (ours) | ✓ | $8^3 + 1024$ | **2.35** | 2.7 | 156 |
| U-Net | ✓ | $48^3$ | 3.33 | 0.5 | 35 |
| FNO | ✓ | $48^3$ | 3.29 | 1.6 | 64 |
| GINO | ✓ | $48^3$ | 2.57 | 7.9 | 360 |
| UPT (ours) | ✓ | $8^3 + 1024$ | **2.25** | 2.7 | 156 |
| U-Net | ✓ | $64^3$ | 2.83 | 1.3 | 86 |
| FNO | ✓ | $64^3$ | 3.26 | 3.8 | 148 |
| GINO | ✓ | $64^3$ | **2.14** | 19.8 | 900 |
| UPT (ours) | ✓ | $8^3 + 1024$ | 2.24 | 2.7 | 156 |

**Profiling memory and runtime**. We evaluate the memory per sample and runtime per epoch (Table 4) by searching the largest possible batchsize via a binary search. To get the memory consumption per sample, we divide the peak memory consumption by the largest possible batchsize. For the runtime per epoch, we conduct a short benchmark run with the largest possible batchsize take and extrapolate the time to 1 epoch. All benchmarks are conducted on a A100 80GB SXM GPU.

## D.5 Transient flows

**Dataset**. The dataset consists of 10K simulations which we split into 8K training simulations, 1K validation simulations and 1K test simulations. Each simulations has a length of 100 seconds where a $\Delta t$ of 1s is used to train neural surrogate models. Note that this $\Delta t$ is different from the $\Delta t$ used by the numerical solver, which requires a much lower $\Delta t$ to remain stable. Concretely, we set the temporal update of the numerical solver initially to $\Delta t = 0.05$s. If instabilities occur, the trajectory is rerun with smaller $\Delta t$. Overall, each trajectory comprises 2K timesteps at the coarsest $\Delta t$ setting and 200K at the finest ($\Delta t = 0.0005$s). The trajectories are randomly generated 2D windtunnel simulations with 1-4 objects (circles of varying size) placed randomly in the tunnel. The uni-directional inflow velocity varies between 0.01 to 0.06 m/s. Each simulation contains between 29K and 59K mesh points where each point has three features: pressure, and the $x-$ and $y-$component of the velocity. In total, the dataset is converted to float16 precision to save storage and amounts to roughly 235GB.

From all training samples, data statistics are extracted to normalize inputs to approximately mean 0 and standard deviation 1. In order to get robust statistics, the normalization parameters are calculated only from values within the inter-quartile range. As the Pytorch function to calculate quartiles is not supported to handle such a large amount of values, we assume that values follow a Gaussian distribution with mean and variance from the data, which allows us to infer the inter-quartile range from the Gaussian CDF.

Additionally, we convert the normalized values $\tilde{u}_{i,k}^t$ to log scale to avoid extreme outliers which can lead to training instabilities:

$$u_{i,k}^t = \text{sign}(\tilde{u}_{i,k}^t) \cdot \log(1 + |\tilde{u}_{i,k}^t|) , \tag{26}$$

where all functions/operations are applied point-wise to the individual vector components. We apply the same log-scale conversion to the decoder output $\hat{u}_{i,k}^t$.

**Model scaling**. We scale models by scaling the dimension of the model. For grid-based methods, we choose 64x64 as grid size which results in similar compute costs of GINO and UPT. As encoder and decoder operate on the mesh — which is more expensive than operating in the latent space — we use half of the approximator hidden dimension as hidden dimension for encoder and decoder. For example, for UPT-68M, we use hidden dimension 384 for the approximator and 192 for encoder/decoder. The hidden dimensions of the approximator are as follows: 128 for UPT-8M, 192 for UPT-17M and 384 for UPT-68M.

For UPT, we use 12 transformer blocks in total, which we distribute evenly across encoder, approximator and decoder (i.e. 4 transformer blocks for each component). As the decoder directly follows the approximator, this implementation detail only matters when training with inverse encoding and decoding to enable a latent rollout. In this setting, the decoder needs more expressive power than a single perceiver layer to decouple the latent propagation and the inverse encoding and decoding.

For GINO, we use 10/12/16 Fourier modes per dimension with a hidden dimension of 40/50/76 for the 8M/17M/68M parameter models, respectively. The hidden dimension of the encoder/decoder are the same ones as used for the UPT models.

For U-Nets, we use the Unet$_{\text{mod}}$ architecture from Gupta & Brandstetter [31] where we adjust the number of hidden dimensions to be approximately equal to the desired parameter counts. We use hidden dimensions 13/21/42 for the 8M/17M/68M parameter models, respectively.

**Implementation**. To efficiently use multi-GPU setups, we sample the same number of mesh points for each GPU. Without this modification, the FLOPS in the encoder would fluctuate drastically per GPU, which results in "busy waiting" time for GPUs with less FLOPS in the encoder. We also keep the number of mesh points constant between batches to avoid memory fluctuations, which allows using a larger batchsize. We rescale the x positions from the range $[-0.5, 0.5]$ to $[0, 200]$ and the y positions from $[-0.5, 1]$ to $[0, 300]$. We do this to avoid complications with the positional embedding [106] which was designed for positive integer positions. To create a graph out of mesh points, we use a radius graph with $r = 5$ and limit the maximum number of graph edges per node to 32 to avoid memory fluctuations. We choose $r = 5$ as it covers the whole domain when using a grid resolution of $64^2$.

**Training**. We train all models for 100 epochs using a batchsize of 1024 where we use gradient accumulation if the full batchsize does not fit into memory. Following common practices, we use AdamW [41, 59] for U-Net, FNO and GINO. As we encountered training instabilities when training UPTs on this dataset, we change the optimizer for UPTs to Lion [21]. These instabilities manifest in sudden loss spikes from which the model can not recover. Using the Lion optimizer together with float32 precision training solves these instabilities in our experiments. When training with Lion, we use a learning rate of $5 \times 10^{-5}$ and $1 \times 10^{-4}$ otherwise. We also use float32 for GINO as we found that this improves performance. U-Net and FNO do not benefit from float32 and are therefore trained in bfloat16 precision.

Other hyperparameters are chosen based on common practices when training ViTs [26]. We linearly increase the learning rate for the first 10 epochs [58] followed by a cosine decay. A weight decay of 0.05 is used when training with AdamW, which is increased to 0.5 when using Lion as suggested in [21]. Architectural choices like the number of attention heads, the latent dimension for each attention head or the expansion dimension for the transformer's MLP are copied from ViTs.

We find float32 precision to be beneficial for GINO and UPTs, which is likely due to positional embeddings being too inaccurate with bfloat16 precision. Training with float16 instead of bfloat16 (to increase the precision of the positional embedding while speeding up training) resulted in NaN values due to overflows. Therefore, we train all GINO and UPT models with float32. We found this to be very important as otherwise UPTs become unstable over the course of training, leading to a large loss spike from which the model can not recover. During model development, we tried various ways to solve this instabilities where using the Lion optimizer was one of them. With fp32 training, UPTs

work similarly well with AdamW but we kept Lion as we already trained models and re-training all models is quite expensive.

**Interpolated models**. We train U-Net and FNO models as baseline on this task. To project the mesh onto a regular grid, we use a $k$-NN interpolation which takes the $k$ nearest neighbors of a grid position and interpolates between the values of the nerarest neighbors. Going from the grid back to the mesh is implemented via the `grid_sample` function of pytorch. For FNO, we use the same dimensions as in latent model of GINO and for U-Net, we scale the latent dimensions such that the model is in the same parameter count range as the other models.

### D.5.1 Visualization of rollouts

We show more qualitative results of the UPT-68M model in Fig. 13.

### D.5.2 Supernode discretization convergence

While one would ideally use lots of supernodes also during training, it also requires more computational power. Therefore, we investigate increasing the number of supernodes only during inference. We use the 68M parameter models from Sec. 4.2 and increase the number of supernodes during inference. Figure 8 shows that UPTs are discretization convergent and inference performance can be improved by using more supernodes for inference than during training.

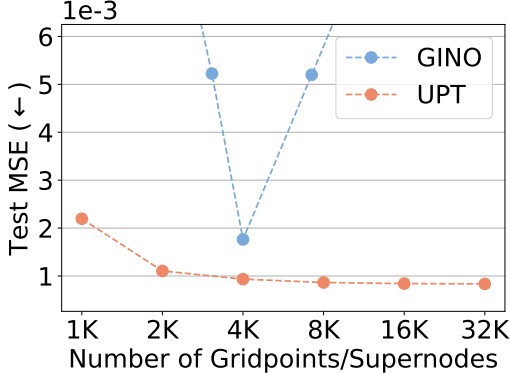

Figure 8: We study discretization convergence by varying the number the number of gridpoints/-supernodes of models that were trained 2K supernodes (UPT) or 4K grid points (GINO). UPT demonstrates a stable performance across different number of supernodes even though it has never seen that number of supernodes during training. Increasing the number of supernodes even improves the performance of UPT slightly. In contrast, the performance of GINO plummets when the number of gridpoints is different during inference.

### D.5.3 Impact of a larger latent space

As training with larger latent spaces becomes expensive, we investigate it in a reduced setting where we train for only 10 epochs and fix the number of input points to 16K. The results in Fig. 9 show that UPTs scale well with larger latent spaces, allowing a flexible compute-performance tradeoff.

### D.5.4 Performance of smaller models

Figure 5 could suggest that UPT does not scale well with parameter size due to the steeper decline of GINO. However, the scaling of GINO only looks good because GINO underperforms in contrast to UPT (GINO-68M is worse than UPT-8M). For well trained (UPT) models it gets increasingly difficult to improve the loss further. Ideally, one would show this by training larger GINO models, however this is prohibitively expensive (68M parameter models already take 450 A100 hours per model). We therefore go in the other directions and train even smaller UPT models that achieve a similar loss to the GINO models and compare scaling there. In Figure 10 right, we compare UPT 1M/2M/8M against GINO 8M/17M/68M. UPT shows similar scaling on that loss scale.

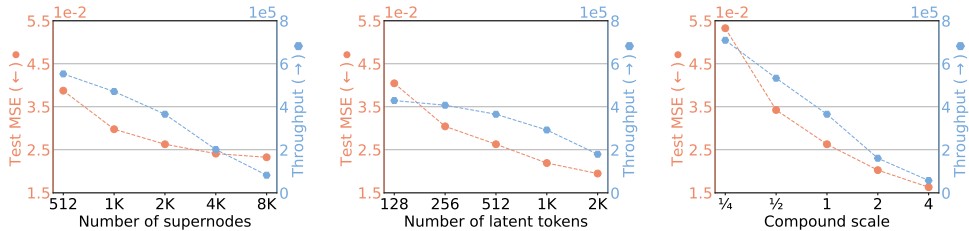

Figure 9: Latent space scaling investigations of a 17M parameter UPT model for 10 epochs. Compound scaling scales the number of supernodes and latent tokens simulataneously where a compund scale of 1 uses $n_{\text{supernodes}}$=2048 and $n_{\text{latent}}$=512, i.e. compound scale 2 uses $n_{\text{supernodes}}$=4096 and $n_{\text{latent}}$=1024. Throughput is measured as number of samples processed per GPU-hour. Models are trained in a reduced setting with 10 epochs and 16K input points.

We strongly hypothesis that the effect of larger UPT models would become apparent in even more challenging settings or even larger datasets. However, challenging large-scale datasets are hard to come by, which is why we created one ourselves. Creating even larger and more complex ones is beyond the scope of our work as it exceeds our current resource budget, but it is definitely an interesting direction for future research/applications.

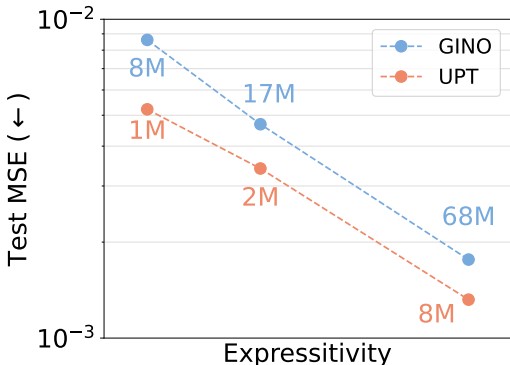

Figure 10: UPT is much more expressive than GINO and shows good scaling when increasing model size. UPT-1M achieves comparable performance to GINO-17M using 17x less parameters.

### D.5.5 Scalability with dataset size

We show the scalability and data efficiency of UPTs by training UPT-8M on subsets of the data used for the transient flow experiments (we train on 2K and 4K out of the 8K train simulations). The results in Figure 11 right show that UPT scales well with data and is data efficient, achieving comparable results to GINO-8M with 4x less data.

### D.5.6 Out-of-distribution generalization study

We study generalization to out-of-distribution datasets by evaluating the 68M parameter models that were trained on the transient flow dataset from Section 4.2. The in-distribution dataset uses an adaptive meshing algorithm, i.e. the mesh resolution around objects is increased, leading to mesh cells of different sizes and contains between 1 and 4 circles of variable size at random locations.

We evaluate three different settings of variable difficulty: more objects, higher inflow velocity. For each setting, we generate a new dataset containing 100 simulations and evaluate the model that was trained on the in-distribution dataset. The results in Figure 12 show that UPTs behave similar to other models when evaluated on OOD tasks where UPT outperforms all other models in all tasks. Therefore, the grid-indpendent architecture of UPTs does not impair OOD generalization performance.

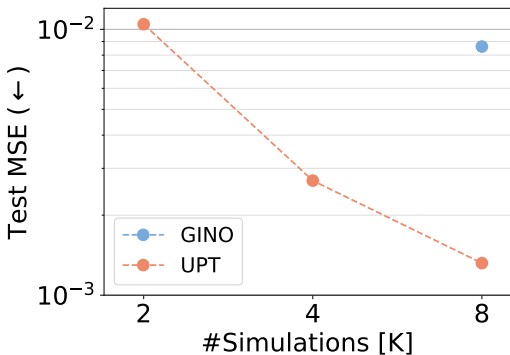

Figure 11: UPT scales well with more data and is data efficient, achieving comparable results to competitors with 4x less data.

**Generalization to more objects**.   Adding additional obstacles to a flow makes it more turbulant. The left side of Fig. 12 shows that also for in-distribution, more objects correspond to higher loss values due to the increased difficulty. Models show stable generalization to larger object counts. Note that we do not encode the number of objects or the objects explicitly in the model. The model simply does not get any inputs in the locations where obstacles are. Therefore, only the dynamics of the flow change but not the distribution of input positions.

**Generalization to higher velocities**.   The in-distribution dataset contains random velocities sampled from $v \in [0.01, 0.06]$ m/s. The velocity is used (together with the timestep) as input to the modulation layers (e.g. DiT [78] modulation) which modulate the features of *all* layers in the network. This leads to OOD velocities distorting *all* features within a forward pass. The center plot of Fig. 12 shows that UPT has the best OOD generalization among all models. Notably, the performance of GINO drastically drops for higher velocities.

**Generalization to different geometries**.   The in-distribution dataset contains randomly placed circles of varying size where the mesh is generated via an adaptive meshing algorithm. To investigate robustness to the meshing algorithm, we generate a OOD dataset with a uniform meshing algorithm. In this dataset, mesh cells are approximately uniform, i.ethe distance between two points is roughly the same. This is in contrast to an adaptive mesh, where regions around an object have more mesh cells and therefore the distance between two points is smaller in these regions. Additionally, we investigate generalization to different obstacle geometries by using polygons (with up to 9 edges) or triangles instead of circles. Note that even though polygons are a more "complicated" shape, they are more reminiscent of a circle than triangles. The size of the obstacle is also varied here. For simplicity, the number of objects per simulation is set to 1 in this study. Note that U-Net and FNO interpolate the mesh onto a 2D grid and therefore their distribution of input positions does not change here, only the dynamics of the simulation. For GINO and UPT, also the distribution of input position changes. The right side of Fig. 12 shows that UPT achieves the best performances on OOD meshes, despite having to adjust to a different input position distribution.

### D.6   Lagrangian fluid mechanics

**Datasets**.   We use the Taylor-Green vortex datasets from Toshev et al. [101] for our Lagrangian experiments, comprising two dimensions (TGV2D) and three dimensions (TGV3D). The Taylor-Green vortex problem is characterized by a distinctive initial velocity field without an external driving force, resulting in a static decrease in kinetic energy over time. The TGV2D dataset includes 2500 particles, a sequence length of 126, and is characterized by a Reynolds number (Re) of 100. The data is split into 100/50/50 for training, validation, and testing, respectively. The TGV3D dataset consists of 8000 particles, a sequence length of 61, and a Reynolds number of 50. The data is partitioned into 200/100/100 for training, validation, and testing, respectively.

**Baselines**.   The Graph Network-based Simulator (GNS) model [87] is a popular learned surrogate for physical particle-based simulations. The architecture is kept simple, adhering to the encoder-

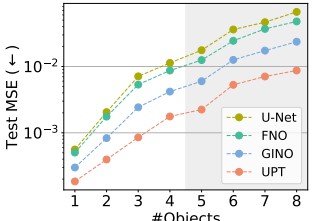 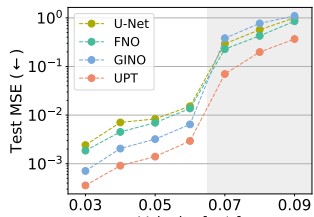 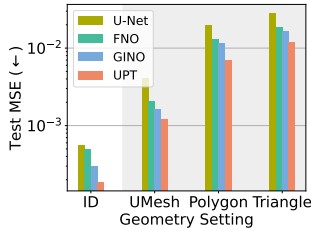

Figure 12: OOD generalization study. Trained 68M parameter models are evaluated on OOD datasets with more objects, higher inflow velocities and different geometries. "ID" refers to an adaptive meshing algorithm with circles as obstacles. "UMesh" changes the meshing algorithm from adaptive to uniform. "Triangle" uses triangles instead of circles and "Polygon" uses polygon obstacles (with up to 9 edges) instead of circles. UPTs have similar OOD generalization capabilities as other models, outperforming them in all evaluations. Grey indicates OOD settings.

processor-decoder principle [6], where the processor consists of multiple graph network blocks [5]. The Steerable E(3)-equivariant Graph Neural Network (SEGNN) architecture [15] is a general implementation of an E(3) equivariant GNN, where layers are directly conditioned on steerable attributes for both nodes and edges. The main building blocks are steerable MLPs, i.e., stacks of learnable linear Clebsch-Gordan tensor products interleaved with gated non-linearities [109]. SEGNN layers are message-passing layers [28] where steerable MLPs replace the traditional non-equivariant MLPs for both message and node update functions.

We utilize the checkpoints provided by Toshev et al. [101] for the comparisons in 4.3. Both GNS and SEGNN baseline models comprise 10 layers with latent dimensions of 128 and 64, respectively. The maximum irreps order of SEGNN are $l = 1$, for more details see [101].

**Hyperparameters**. We train using a batchsize of 128 for 50 epochs with a warmup phase of 10 epochs. We sample inputs between $50\%$ to $100\%$ of total particles. For optimization, we use AdamW [59] for all experiments with learning rate $10^{-3}$ and weight decay of 0.05.

**UPT architecture**. For the TGV2D and TGV3D experiments, we use $n_s = 256$ and $n_s = 512$ supernodes, respectively. For TGV3D more supernodes are required due to the increased number of particles. Message passing is done with a maximum of four input points per supernode. The flexibility of the UPT encoder allows us to randomly sample $50\%$ up to $100\%$ of the total particles, diversifying the training procedure and forcing the model to learn the underlying dynamics.

The encoder comprises four transformer blocks with a latent dimension of 96 and two attention heads. The encoder perceiver expands the latent dimension to 192 and outputs 128 latent tokens, using 128 learned queries and three attention heads. The approximator comprises four transformer blocks with a latent dimension of 192 and three attention heads. The decoder perceiver comprises again three attention heads. Overall, the parameter count of UPT amounts to 12.6M parameters.

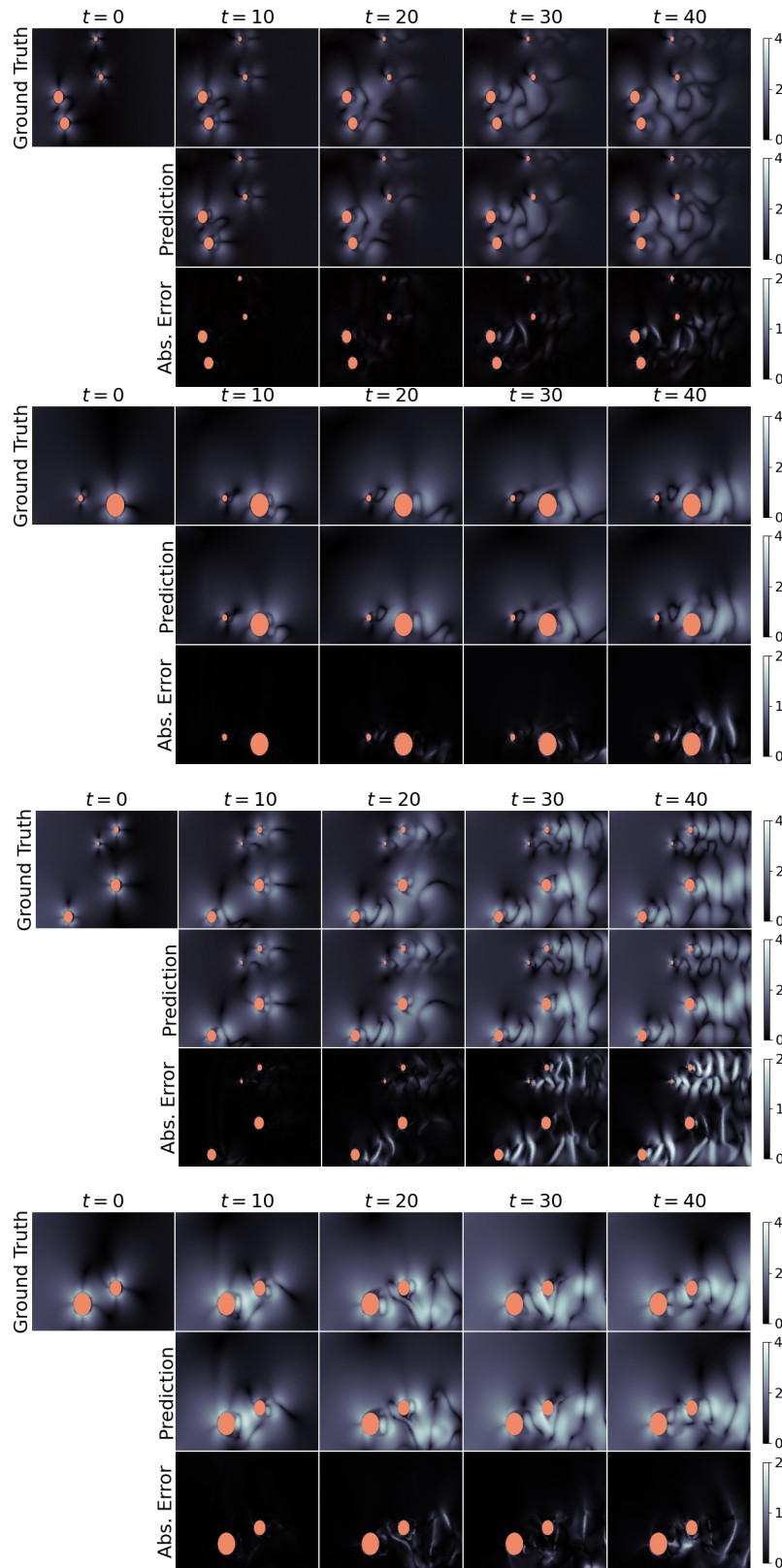

Figure 13: Exemplary visualizations for transient flow simulation rollouts. Best viewed zoomed in.

**Experiments on Taylor Green Vortex 2D**.

We show results on the TGV2D dataset from LagrangeBench [101] and the time to generate a simulated trajectory in Figure 14. The velocity error behaves similar to the 3D version of the dataset in Figure 7. UPT is roughly 10 times faster than GNNs and 100 times faster than the classical SPH solver.

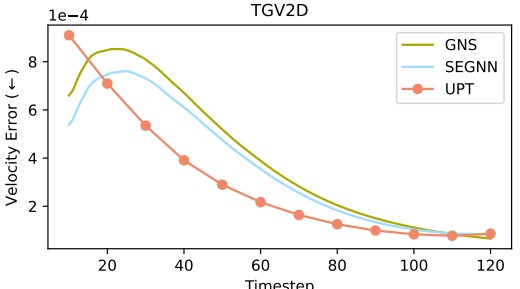

| Model | Time | Speedup | Device |
|-------|------|---------|--------|
| SPH | 4.9s | 1x | A6000 |
| SEGNN | 2.76s | 2x | A100 |
| GNS | 0.56s | 9x | A100 |
| UPT | **0.05s** | **98x** | A100 |

Figure 14: Left: Mean Euclidean norm of the velocity error over all particles for different timesteps. UPTs effectively learn the underlying field dynamics, resulting in lower velocity error as the trajectory evolves in time. Right: Comparison of simulation/rollout runtimes for a TGV2D trajectory with 125 timesteps and 2500 particles across SPH simulation.

## D.7 Qualitative study of scaling limits details

We qualitatively study the scaling limits of representative methods by evaluating the memory consumption when training a 68M parameter model with a batchsize of 1 while scaling the number of inputs in the left part of Fig. 2.

We study the following models:

- GNN: Lagrangian based simulations are currently dominated by GNN architectures due to their strong ability to update individual particles based on representation of neighboring particles. In this study, we consider a GNO (i.e. a GNN with radius graph and mean aggregation) since edge information is typically absent in physical simulations, but this does not change the asymptotic complexity. Additionally, we limit the number of edges per node (degree of the graph) to 4 as otherwise the GNN could not fit 32K inputs on 80 GB of GPU memory. Note that this is a heavy restriction and these models would most likely need a larger degree to perform well when actually trained.

- Transformer: The amount of operations required for the vanilla transformer self-attention is quadratic in the number of inputs and therefore becomes increasingly expensive when the number of input grows. Linear transformers change the attention mechanism to have linear complexity and therefore scale better to larger input spaces. We use the linear attention implementation of [90], but note that the complexity is equivalent to recent transformer architectures proposed as neural surrogate for partial differential equations [18, 53, 33].

- GINO: Aggregating information into a lower dimensional latent space allows models to be scaled to a large number of inputs. GINO is one of the prominent models that operates with this paradigm by projecting the input point cloud onto points on a regular grid.

- UPTs: The efficient latent space compression of UPTs allows scaling to large number of input points.

We do not include FNO or CNN/U-Net into this study as they can only handle inputs from a regular grid. Therefore, they would need a very fine-grained grid to achieve comparable results to GINO or UPTs, which makes a fair comparison hard. Note that GINO usea a FNO after it projects inputs onto a regular grid. Therefore, the scalability of GINO is highly correlated with the scalability of FNOs or CNN/U-Nets (i.e. decent scalability on 2D problems but drastically higher compute requirements on 3D problems). We also exclude transformers with quadratic attention as they quickly become infeasible and their linear complexity counterpart scales linear with the number of inputs.

We also scale the latent space size for GINO (number of gridpoints) and UPTs (number of supernodes and number of latent tokens). As starting point, we use the latent space sizes used in Sec 4.2. For GINO-3D, we use the same grid resolution for the 3rd dimension as for GINO-2D (i.e. GINO-2D uses $64^2$ and GINO-3D $64^3$). Our experiments in Sec. 4.1 and Sec. 4.2 show that GINO needs this high resolution for the grid to achieve good performances. For GINO-3D we set the number of fourier modes to $5$ to keep the parameter count close to 68M.

As a quantitative comparison between completely different architectures is challenging, we additionally list the theoretical model complexities in Tab. 5. This study is meant to give a qualitative impression of the practical implications of the theoretical asymptotic complexities.

Table 5: Extending the right table of Fig. 2 with theoretical asymptotic complexities. Complexity includes number of mesh points $M$, and maximum degree of the graph $D$. Grid-based methods project the mesh to $G$ grid points. UPTs instead use a small amount of supernodes $S$ as discretization, where $G$ is typically much larger than $S$. The UPT training procedure separates responsibilities between components, allowing us to forward propage dynamics purely within the latent space.

| Model | Range | Complexity | Irregular Grid | Discretization Convergent | Learns Field | Latent Rollout |
|---|---|---|---|---|---|---|
| GNN | local | $O(MD)$ | ✓ | ✗ | ✗ | ✗ |
| CNN | local | $O(G)$ | ✗ | ✗ | ✗ | ✗ |
| Transformer | global | $O(M^2)$ | ✓ | ✓ | ✗ | ✗ |
| Linear Transformer | global | $O(M)$ | ✓ | ✓ | ✗ | ✗ |
| GNO [52] | radius | $O(MD)$ | ✓ | ✓ | ✗ | ✗ |
| FNO [51] | global | $O(G \log G)$ | ✗ | ✓ | ✗ | ✗ |
| GINO [54] | global | $O(GD + G \log G)$ | ✓ | ✓ | ✓ | ✗ |
| UPT | global | $O(SD + S^2)$ | ✓ | ✓ | ✓ | ✓ |

# E  Impact Statement

Neural PDE surrogates play an important role in modeling many natural phenomena, many of which are related to computational fluid dynamics. Examples are weather modeling, aerodynamics, biological engineering, or plasma physics. Given the widespread application of computational fluid dynamics, obtaining shortcuts or alternatives for computationally expensive simulations, is essential for advancing scientific research, and has direct or indirect implications for reducing carbon emissions. In contrast to traditional computational fluid dynamics simulations, our approach aims to eliminate the need for complex meshing, bypass the requirement for detailed knowledge of boundary conditions, and overcome bottlenecks caused by solver intricacies. However, it is important to note that relying on simulations always necessitates thorough cross-checks and monitoring, especially when employing a "learning to simulate" methodology.

# F  Implementation Details

## F.1  Supernode pooling

Supernodes are, in the case of Eulerian data, sampled according to the underlying mesh and, in the case of Lagrangian data, sampled according to the particle density. A random sampling procedure which follows the mesh or particle density, respectively, allows us to put domain knowledge into the architecture. Consequently, complex regions are accurately captured, as these regions will be assigned more supernodes than regions with a low-resolution mesh or few particles.

The sampling of supernodes is done for each optimization step which provides regularization.

The radius graph encodes a fixed region around each supernode. We do not use any information (such as cell connectivity/adjacency) from the original mesh in the Eulerian case. While one could use the original edge connections for Eulerian data, Lagrangian data does not have edges. Additionally, we employ randomly dropping input nodes as a form of data augmentation which makes using the original edge connections more complicated from an implementation standpoint. In contrast, our design via radius graph is agnostic to the simulation type and also to randomly dropping input nodes.

We choose the radius depending on the dataset. For each dataset, we first analyze the average degree of the radius graph with different radius values. We then choose the radius such that the degree is around 24, i.e., on average each supernode represents 24 inputs. We found our model to be fairly robust to the radius choice. Also, the circles of different supernodes can overlap. Therefore, the encoding of dense regions can be distributed among multiple supernodes.

We impose the edge limit of the radius graph by randomly dropping connections to preserve the average distance from supernode to input nodes and avoid "squashing" too much information into a single supernode. We choose this edge limit to be 32 in all cases.

### F.2   Position embedding

We embed input and query positions via the sine-cosine position embedding from transformers [106]. Each position dimension is embedded separately. We rescale all positions to be in the range [0, 200] to avoid "unsmooth" transitions when going from positive positions to negative positions. This also makes sure that positions are in a suitable scale for the transformer positional embedding. For example, input positions could be arbitrarily small, which would lead all positions to be the same due the limited precision of float values. The upper bound 200 is a hyperparameter where a broad range of values works. If input nodes have additional input features, these features are first linearly projected into a hidden dimension and then the positional embedding vector is added.

We also experimented with random fourier features [96], but did not find a significant difference.

### F.3   Conditioning pseudocode

We provide pseudocode how to create a conditioning vector to encode scalar properties into the model. This conditioning vector is then used to predict the parameters of the LayerNorm [2] layers before the attention and MLP parts of each transformer block. Additionally, a third scalar for gating the residual connections is learned. This methodology follows DiT [78]. For predicting these scalars, a simple learnable linear projection is used.

```
1  # embed: sine - cosine positional embedding
2  # condition_mlp: shallow MLP to combine boundary conditions
3  def create_condition(timestep, velocity):
4      timestep_embed = embed(timestep)
5      velocity_embed = embed(velocity)
6      embed = concat([timestep_embed, velocity_embed])
7      condition = condition_mlp(embed)
8      return condition
```
Listing 1: PyTorch-style pseudocode for conditioning onto scalar variables.

### F.4   Training pseudocode

We provide pseudocode for a training step in the setting of the transient flow experiments (2D positions with 3 features per node) in Listing 2 including inverse encoding/decoding objectives.

```python
1   # input_embed: linear projection from input features dim to hidden dim
2   # pos_embed: sine-cosine positional embedding
3   # message_mlp: shallow MLP to create messages
4   # encoder_transformer: stack of transformer blocks of the encoder
5   # latent_queries: 'n_latent_tokens' learnable query vectors
6   # encoder_perceiver: cross attention block
7   # approximator_transformer: stack of transformer blocks
8   # query_mlp: shallow MLP in the decoder to encode query positions
9   # decoder: cross attention block
10
11  def encoder(input_features, input_pos, radius, n_supernodes):
12      """
13      encode arbitrary pointclouds into a fixed latent space
14      inputs:
15          input_features Tensor(n_input_nodes, 3): features of input
    nodes
16          input_pos Tensor(n_input_nodes, 2): positions of input nodes
17          n_supernodes integer: number of supernodes
18          radius float: radius for creating the radius_graph
19      outputs:
20          latent Tensor(n_latent_tokens, hidden_dim): encoded latent
21      """
22      # create radius graph (using all input nodes)
23      # edges are uni-directional
24      # messages are passed from nodes_from to nodes_to
25      nodes_from, nodes_to = radius_graph(input_pos, radius)
26
27      # select supernodes from input_nodes
28      n_input_nodes = len(input_features)
29      supernode_idx = randperm(n_input_nodes)[:n_supernodes]
30
31      # filter out edges that do not involve supernodes
32      is_supernode_edge = nodes_to in supernode_idx
33      nodes_from = nodes_from[is_supernode_edge]
34      nodes_to = nodes_to[is_supernode_edge]
35
36      # encode inputs and positions
37      encoded_nodes = input_embed(input_features) + pos_embed(input_pos)
38
39      # create messages
40      messages = message_mlp(encoded_nodes[nodes_from])
41      # accumulate messages per supernode by averaging messages
42      supernodes = accumulate(messages, nodes_to, reduce="mean")
43
44      # process supernodes with some transformer blocks
45      supernodes = encoder_transformer(supernodes)
46      # perceiver pooling from supernodes to latent tokens
47      latent = encoder_perceiver(
48          query=latent_queryies,
49          key=supernodes,
50          value=supernodes,
51      )
52      return latent
53
54  def approximator(latent):
55      """
56      propagates latent forward in time
57      inputs:
58          latent_t Tensor(n_latent_tokens, hidden_dim):
59              encoded latent space at timestep t
60      outputs:
61          latent_t_plus_1 Tensor(n_latent_tokens, hidden_dim):
62              encoded latent space at timestep t + 1
63      """
64      return approximator_transformer(latent)
```

```
65
66  def decoder(latent, query_pos):
67      """
68      decode latent space pointwise at arbitrary positions
69      inputs:
70          latent Tensor(n_latent_tokens, hidden_dim): latent space
71          query_pos Tensor(n_outputs, 2):
72              positions for querying the latent space
73      outputs:
74          decoded Tensor(n_outputs, 3):
75              evaluation of the latent space at query positions
76      """
77      # encode query positions
78      query_pos_embed = query_mlp(pos_embed(query_pos))
79      # query latent space
80      decoded = decoder(query=query_pos_embed, key=latent, value=latent)
81      return decoded
82
83  def step(input_features, input_pos, n_supernodes, radius, query_pos):
84      """
85      encode arbitrary pointclouds into a fixed latent space
86      inputs:
87          input_features Tensor(n_input_nodes, 3):
88              features of input nodes at timestep t
89          input_pos Tensor(n_input_nodes, 2):
90              positions of input nodes at timestep t
91          n_supernodes integer: number of supernodes
92          radius float: radius for creating the radius_graph
93          query_pos Tensor(n_outputs, 2):
94              positions for querying the latent space
95          target_features Tensor(n_input_nodes, 3):
96              features of nodes at timestep t + 1
97      outputs:
98          loss Tensor(1): skalar loss value
99      """
100     # next-step prediction
101     latent_t = encoder(
102         input_features,
103         input_pos,
104         radius,
105         n_supernodes,
106     )
107     latent_tplus1 = approximator(latent_t)
108     decoded_tplus1 = decoder(latent_tplus1, query_pos)
109     next_step_loss = mse(decoded_tplus1, target_features)
110     # inverse decoder (decode latent into inputs)
111     decoded_t = decoder(latent_t, input_pos)
112     inverse_decoding_loss = mse(decoded_t, input_features)
113     # inverse encoder (encode predictions at t+1 into latent of t+1)
114     inverse_encoded = encoder(
115         decoded_tplus1,
116         query_pos,
117         radius,
118         n_supernodes,
119     )
120     inverse_encoding_loss = mse(inverse_encoded, latent_tplus1)
121     # sum losses
122     return (
123         next_step_loss
124         + inverse_decoding_loss
125         + inverse_encoding_loss
126     )
```

Listing 2: PyTorch-style pseudocode for UPT with inverse encoding an decoding objecitves.

