# OpenReview forum: "Universal Physics Transformers: A Framework For Efficiently Scaling Neural Operators"
_NeurIPS.cc/2024/Conference — NeurIPS 2024 poster_

### Official Review · Reviewer_1stF · 2024-07-01

**Soundness:** 3
**Presentation:** 3
**Contribution:** 3
**Rating:** 7
**Confidence:** 3

**Summary:**

The authors presented Universal Physics Transformers (UPTs), a framework for efficient learning and scaling neural operators. UPTs offer flexibility in handling various data types, whether grid-based or particle-based. UPTs compress data into a low-dimensional latent space and perform dynamics propagation within this space, resulting in fast simulations. Additionally, the latent representation allows evaluation at any point in space-time. UPTs demonstrated strong performance when tested on various fluid dynamics problems.

**Strengths:**

- The authors developed a model based on linear attention that handles both mesh-based and particle-based data effectively.
- It leverages a low-dimensional latent space for efficient temporal dynamics propagation, enabling rapid evaluations.
- They conducted very interesting Navier-Stokes experiments.
- They demonstrated that UPT serves as a robust and efficient baseline.
- In scalability experiments, UPT showed excellent performance, scaling efficiently with input size and outperforming other baselines across all tested scales.
- They examined the model's convergence concerning the number of I/O points, obtaining good outcomes.

**Weaknesses:**

- The framework is labeled as universal but is only tested on fluid dynamics, lacking experiments on other PDEs.
- Although the model scales well with input size, its scalability with model size appears poor based on a conducted experiment (Figure 5). While it outperforms other baselines, its scaling rate is significantly lower than that of models like GINO. Additional experiments are necessary to address this issue (if the UPT is meant to be used in alrge scale applications)
- The scalability of UPT concerning training size remains unclear, which is an important property.
- The stability of the latent rollout technique for long rollouts is uncertain, raising concerns about its general reliability

**Questions:**

- What do you think causes the performance to plateau with increasing model size as observed in Figure 5? Is it because the error is already sufficiently low, preventing further enhancement?
- Why do you believe that latent rollouts do not require stabilization?

**Limitations:**

The authors explained well the limitations.

---

> ### Author Rebuttal · Authors · 2024-08-07
>
> We appreciate your review and respond to the raised concerns below.
>
> **Experiments with other PDEs**
>
> UPTs are neural operators known to be applicable across various PDE types. Due to the nonlinear nature of the NS equations, they are notoriously more challenging to solve than parabolic or hyperbolic PDEs, like the heat or wave equations. Therefore, we hypothesize that UPTs should also generalize well to different PDEs.
>
>
> Additionally, as requested by reviewer JKEE, we added comparisons of UPT with other methods on small-scale regular grid datasets. In total, UPT outperforms competitors on 5 diverse datasets that span over different dataset sizes (900 to 1M frames), resolutions (2K to 60K inputs), spatial dimensions (2D, 3D), simulation types (steady state, transient), different boundary conditions and specifications (Eulerian, Lagrangian).
>
>
>
> **Scalability with model size**
>
>
> We see now that Figure 5 could suggest that UPT does not scale well with parameter size. However, the scaling of GINO only looks good because GINO underperforms in contrast to UPT (GINO-68M is worse than UPT-8M). As you correctly identified, for well trained (UPT) models it gets increasingly difficult to improve the loss further. Ideally, one would show this by training larger GINO models, however this is *prohibitively* expensive (68M parameter models already take 450 A100 hours per model). We therefore go in the other directions and train even smaller UPT models that achieve a similar loss to the GINO models and compare scaling there. In Figure 2 of the supplemental pdf, we compare UPT 1M/2M/8M against GINO 8M/17M/68M. UPT shows similar scaling on that loss scale.
>
> We see how this could be easily misinterpreted from Figure 5 and adjust the paper accordingly to remove this misunderstanding.
>
> We strongly hypothesis that the effect of larger UPT models would become apparent in even more challenging settings or even
> larger datasets. However, challenging large-scale datasets are hard to come by, which is why we created one ourselves. Creating even larger and more complex ones is beyond the scope of our work as it exceeds our current resource budget, but it is definitely an interesting direction for future research/applications.
>
>
>
>
>
>
>
>
>
>
>
>
>
>
> **Scalability with dataset size**
>
> We added experiments to show the scalability and data efficiency of UPTs by training UPT-8M on subsets of the data used for the transient flow experiments (we train on 2K and 4K out of the 8K train simulations). The results in Figure 3 of the supplementary rebuttal pdf show that UPT scales well with data and is data efficient, achieving comparable results to GINO-8M with 4x less data.
>
> We also show that UPTs can handle various dataset sizes. ShapeNet-Car is a small-scale dataset consisting of 889 car shapes with 3.6K mesh points each. TGV3D is a bit bigger with 8K particles per simulation and 200 simulations of length 61 (12K frames). The dataset for our transient flow experiments contains around 50K mesh cells per simulation with 10K simulations of length 100 (1M frames).
> UPT shows strong performances across all considered dataset sizes.
>
>
> Additionally, UPT consists of mostly transformer blocks, which have demonstrated outstanding scalability in other domains such as language modeling [1] and computer vision [2].
>
>
> [1] Kaplan et al., "Scaling laws for neural language models", arXiv 2020, https://arxiv.org/abs/2001.08361
>
>
> [2] Zhai et al., "Scaling vision transformers", CVPR 2022, https://arxiv.org/abs/2106.04560
>
>
> **Stability of latent rollout**
>
>
> We found UPT to be fairly stable without any special techniques to stabilize the rollout (e.g. [3]). However, such methods could further improve UPTs performance but these methods are not specific to UPT and typically require additional computations during training. Therefore, we leave exploration of this combination to future work.
>
> Additionally, the latent rollout opens new avenues to potentially apply existing stabilization tricks with less compute. For example, one could do the forward propagation for the stabilization technique from [4] in the latent space, which would greatly reduce training costs thereof. However, as these tricks can be tricky to train (e.g. due to requiring a precise trade-off between training the next-step prediction vs n-step prediction), we leave this direction to future work.
>
>
>
> [3] Lippe et al., "Pde-refiner: Achieving accurate long rollouts with neural pde solvers", NeurIPS 2023, https://arxiv.org/abs/2308.05732
>
> [4] Brandstetter et al., "Message passing neural pde solvers", ICLR 2022, https://arxiv.org/abs/2209.15616

---

> > ### Comment · Reviewer_1stF · 2024-08-08
> >
> > Thank you for your answers. I increased the rate to 7. Good luck!

---

### Official Review · Reviewer_QzDc · 2024-07-02

**Soundness:** 4
**Presentation:** 3
**Contribution:** 3
**Rating:** 6
**Confidence:** 4

**Summary:**

This paper introduces Universal Physics Transformers (UPTs) to provide a unified learning paradigm for grid- or particle-based structures, enabling scalability across meshes and particles. UPTs mainly follow Encode-Process-Decode paradigm and allow queries at any space-time point through perceiver-like cross attention. To separate the responsibilities of individual components, the authors introduce inverse encoding and decoding losses in addition to the next-step prediction loss. Extensive experiments are conducted in diverse applications, including mesh-based fluid simulations and Lagrangian dynamics.

**Strengths:**

UPTs provide a unified framework that can handle both grid- and particle-based structures, enhancing flexibility across different simulation types. UPTs reduce computational overhead by selecting and aggregating features on supernodes within mesh- and particle-based structures. The paper is well-written with a clear structure.

**Weaknesses:**

- Regardless of the structure of unstructured mesh/structured mesh/point clouds, when processing with UPTs, each node is actually modeled as a token, and a transformer architecture is built on this basis. However, there have been several works [1, 2, 3] that utilize transformers or attention mechanisms to model PDE problems. The novelty of using UPTs is somewhat limited in this way.
- The paper misses comparisons to existing work that employs transformer and attention mechanisms for similar purposes.  While [1, 2, 3] have designed methods to reduce the computation overhead of transformer or attention mechanisms, the UPTs approach uses a simple random sampling to aggregate supernode features to reduce computation overhead.
- While the latent rollout approach significantly accelerates inference speed, the experimental results do not show a clear improvement over the autoregressive unrolling via the physics domain. This raises concerns about the effectiveness and practicality of the latent rollout method.

[1] GNOT: A General Neural Operator Transformer for Operator Learning

[2] Transolver: A Fast Transformer Solver for PDEs on General Geometries

[3] Transformer for Partial Differential Equations' Operator Learning

**Questions:**

- Given that the latent rollout approach does not provide a significant improvement in experimental results compared to autoregressive unrolling, can you elaborate on the potential reasons behind this? Specifically, do you believe that the lack of clear separation between the encoder, approximator, and decoder in this training framework contributes to this issue? Additionally, how effective do you find the inverse encoding and decoding processes? Considering the substantial training overhead and marginal performance gains, what are the key motivations for choosing the latent rollout method over other approaches? Are there any foreseeable optimizations or modifications that could enhance its efficacy?
- How dose DiT modulation influences UPTs' performance? Additionally, which features have you found to be most effective as conditions for DiT modulation?
- How do you specifically determine the number of supernodes in each region of mesh-structured data? After sampling nodes on the mesh, the original edge connections are lost. What impact does using a radius graph to replace the original mesh connectivity have on the results? Can you provide details on how this transformation affects the model's performance and accuracy?
- How is the radius for the radius graph determined when aggregating information at the selected supernodes via a message passing layer? Additionally, do the circles of different supernodes overlap?
- For each optimization step, are the supernodes fixed or do they change dynamically?
- It is mentioned that the edge limit is imposed on nodes in a radius graph to prevent memory fluctuations. Is the edge limit satisfied if the radius is smaller than the set value, or if the limit is exceeded, are edges randomly removed to comply with the constraint?

**Limitations:**

Yes

---

> ### Author Rebuttal · Authors · 2024-08-07
>
> Thank you for your review and helpful comments which were very useful to improve the paper. We address your points individually.
>
>
> **Comparison to other transformer methods**
>
>
>
>
>
>
>
> The fundamental building principles of UPT are (i) an encoding that is designed to handle irregular grids of various sizes, (ii) a compressed and unified latent space representation that allows for efficient forward propagation in the latent space, and (iii) a neural field-type decoder. We consider all of these points as fundamental for scaling neural operators.
>
> GNOT processes each mesh point as a token and therefore quadratically scales with the number of mesh points.
> Furthermore the output can only be evaluated at the positions of the input.
>
> Transolver's concept of "Physics-aware Tokens" is somewhat similar to UPTs supernodes.
> However, only the substitutes attention part operates on a reduced space, whereas the FFN part of the transformer operates on the uncompressed space. Therefore, it is a type of linear complexity transformer, which quickly becomes infeasible for larger input sizes (see Figure 2).
> Similarly, OFormer also operates on the uncompressed input.
>
> In contrast, UPT heavily compresses the input, leading to sub-linear complexity w.r.t. input size in all transformer layers.
>
> Additionally, we added comparisons to transformer baselines on regular grid datasets (see general response) where UPT outperforms e.g. OFormer by quite a big margin.
>
>
>
>
>
>
>
>
>
>
>
>
>
> **Benefits of the latent rollout**
>
> While the latent rollout does not provide a significant performance improvement, it is almost an order of magnitude faster.
> The UPT framework allows to trade-off training compute vs inference compute. If inference time is crucial for a given application, one can train UPT with the inverse encoding and decoding objectives, requiring more training compute but greatly speeding up inference. If inference time is not important, one can simply train UPT without the reconstruction objectives to reduce training costs.
>
> Additionally, the latent rollout enables applicability to Lagrangian simulations. As UPT models the underlying field instead of tracking individual particle positions it does not have access to particle locations at inference time. Therefore, autoregressive rollouts are impossible since the encoder requires particle positions as input. Using the latent rollout, it is sufficient to know the initial particle positions as dynamics can be propagated without any spatial positions. After propagating the latent space forward in time, one can simply query the latent space at arbitrary positions to evaluate the underlying field at given positions. We showcase this in Figure 7 where the latent space is queried with regular grid coordinates (white arrows).
>
>
> We discuss a potential improvement for the latent rollout to make it more efficient in Appendix A. While we show in the paper that a latent rollout can be enabled via a simple end-to-end training, we think that it can be improved, e.g. via a two stage procedure of first training encoder/decoder in an autoencoder setting, followed by freezing encoder/decoder and training the approximator on the fixed pre-trained latent space akin to [1]. Such an approach would not require inverse encoding/decoding objectives as the separation of components is enforced through the multi-stage training.
>
> [1] Rombach et al., "High-resolution image synthesis with latent diffusion models", CVPR 2022, https://arxiv.org/abs/2112.10752
>
> **DiT modulation**
>
> Conditioning onto external features is crucial to encode this external information into the model. We condition onto the current timestep and also onto the inflow velocity in the transient flow experiments.
> Note that this is not specific to UPT and we also apply conditioning to compared models.
>
>
>
>
>
>
> **Supernodes and radius graph creation**
>
>
> We realize that our description in the paper is a bit misleading. Supernodes are, in the case of Eulerian data, sampled according to the underlying mesh and, in the case of Lagrangian data, sampled according to the particle density. A random sampling procedure which follows the mesh or particle density, respectively, allows us to put domain knowledge into the architecture. (We change "randomly sampled" to "sampled according to the underlying mesh / underlying particle density".)
> Consequently, complex regions are accurately captured, as these regions will be assigned more supernodes than regions with a low-resolution mesh or few particles.
>
>
>
>
> The sampling of supernodes is done for each optimization step thus having a regularization effect.
>
> The radius graph encodes a fixed region around each supernode. While one could use the original edge connections for Eulerian data, Lagrangian data does not have edges. Additionally, we employ randomly dropping input nodes as a form of data augmentation which makes using the original edge connections more complicated from an implementation standpoint. In contrast, a radius graph is agnostic to randomly dropping input nodes.
>
> We choose the radius depending on the dataset. For each dataset, we first analyze the average degree of the radius graph with different radius values. We then choose the radius such that the degree is around 16, i.e. on average each supernode represents 16 inputs. We found our model to be fairly robust to the radius choice. Also, the circles of different supernodes can overlap. Therefore, the encoding of dense regions can be distributed among multiple supernodes.
>
> We impose the edge limit of the radius graph by randomly dropping connections to preserve the average distance from supernode to input nodes.
>
>
>
>
>
> Additionally, the radius graph facilitates Lagrangian settings where there is no predefined connectivity between particles.
>
>
> We extended discussion in the corresponding sections and added a guide on how to choose these hyperparameters to the paper.

---

> > ### Comment · Reviewer_QzDc · 2024-08-08
> > **Reply to rebuttal**
> >
> > Thank you for the clarifications, and the additional experiments in general response which I believe strengthen the paper. I have raised my score. However, it would be better if the supernode pooling/radius graph construction could be included either in the maintext or supplementary for reference.

---

### Official Review · Reviewer_CUfq · 2024-07-07

**Soundness:** 3
**Presentation:** 3
**Contribution:** 4
**Rating:** 7
**Confidence:** 3

**Summary:**

This paper introduces a transformer-based architecture to scale neural operators to larger and more complex conditions involving spatiotemporal modeling. The novelty of this architecture is the use of the so-called latent rollout, which performs autoregressive modeling on the latent space, as opposed to the decoded space in other approaches. Authors claim that this architecture can operate without spatial structures, which is an inherent benefit of transformer-based approaches, and inspection of the latent-space in space-time. The authors show the usefulness of this approach across different physics-based problems. This is a meaningful contribution to design of NNs for physics applications, with potential broader applications in other ML domains. However, there are some questions that this reviewer hopes that the authors can address.

**Strengths:**

1. Paper is clearly written, and appendices help with self-completeness.
2. Proposed method is shown to be better than existing popular architectures for physics-based modeling.
3. Code is provided for reproducibility.

**Weaknesses:**

1. A bit more analysis and information could be included to improve completeness. See Questions.
2. Only test MSE has been used as an accuracy metric. Physics applications typically care about conservation of mass and performance near boundaries. However, the lack of further metrics is somewhat justified by the speedup and memory gains shown,

**Questions:**

1. Authors should be commended on the discussion on memory complexity in Line 105 to 119. However, this could be summarized as a table for cleaner presentation. Can the authors also include time complexity information for a more transparent comparison of the trade-offs associated with the different architectures? (I know this has been somewhat benchmarked in Table 1, but it could be useful to have some more theoretical information)
2. What kind of position encoding is used for the UPT? This is unclear in the paper. Does the type of position encoding matter for Lagrangian vs Eulerian.
3. For a more comprehensive display of results, could the authors include accuracy metrics (e.g., mean error) and memory benefits to  Table 1?
4. This has been briefly discussed in Appendix A. But for further clarification, is the present architecture approach limited to physics-related applications? Or are there opportunities for extending the present approach towards other autoregressive ML applications, such as video or language modeling?

**Limitations:**

Limitations are well-discussed in Appendix A.

---

> ### Author Rebuttal · Authors · 2024-08-07
>
> Thank you for your comments which helped to improve the paper. We addressed all your comments and followed all your suggestions.
>
> **Memory and time complexity of different architectures**
>
>
> We provide theoretical complexities in Appendix C.6 in Table 3. Note that when scaling only the input size towards infinity to calculate an asymptotic runtime/memory complexity, the runtime complexity is mostly the same as the memory complexity as storing intermediate activation dominates the other factors. For example, transformers scale quadratically with the number of inputs M. For large M, a transformer needs to calculate the MxM attention matrix and also store it in memory. Therefore, it has $O(M^2)$ runtime and memory complexity.
> For simplicity, we only consider the theoretical case and do not take optimizations such as hardware-aware implementations (e.g. [1]) into account that can trade-off additional computations for a reduced memory footprint.
>
> As the theoretical complexities introduce many variables due to different architectures processing the input in vastly different ways, we believe that Figure 2 presents the practical complexities in a cleaner way and provide the theoretical complexities later in the appendix.
>
> [1] Dao et al., "FlashAttention: Fast and Memory-Efficient Exact Attention with IO-Awareness", NeurIPS 2022, https://arxiv.org/abs/2205.14135
>
>
>
> **Positional encoding type**
>
>
>
> We employ the same positional embedding approach used in the original transformer paper [2], applying it separately to each dimension, as is also common in vision transformers [3]. Namely, we employ a combination of sine and cosine functions with different frequencies to encode each position. The revised version of the paper offers a clearer explanation of this positional embedding.
>
> We also experimented with different types of positional embeddings such as the Fourier feature mapping from [4], which uses randomly sampled frequencies from a Gaussian distribution. In an experiment on TGV2D of our Lagrangian experiments it shows a slight improvement (see Figure 1 of the supplemental rebuttal pdf). The results show a slight improvement for longer rollouts, which we hypothesize is because this method doesn't overly emphasize features aligned with the axes. However, as improvements of other positional embeddings were minor, we stuck to the transformer positional embedding for simplicity.
>
>
> [2] Vaswani et al., "Attention is all you need", NeurIPS 2017, https://arxiv.org/abs/1706.03762
>
> [3] Dosovitskiy et al., "An Image is Worth 16x16 Words: Transformers for Image Recognition at Scale", ICLR 2021, https://arxiv.org/abs/2010.11929
>
> [4] Tancik et al., "Fourier features let networks learn high frequency functions in low dimensional domains", NeurIPS 2020, https://arxiv.org/abs/2006.10739
>
>
>
> **Additional accuracy metrics and memory benefits of Table 1**
>
> We included memory benefits in Table 3 of Appendix C.6. However, we find it difficult to assign a representative accuracy metric to each architecture
> as most methods are specialized for a certain type of task. For example, the accuracy of a CNN on a task with irregular data will naturally be worse than the accuracy of a GNN on that problem. UPT shows strong performances across various input domains and input scales, but as other methods are not as flexible we find it difficult to quantify their performance with a single metric.
>
>
> **Application in other domains**
>
> Our method could also be applied to other fields, such as video modeling. In that context, it would be possible to train on a dataset with varying resolutions. Since the decoder's output is a point-wise neural field, it could generate videos at arbitrarily high resolutions.
> However, in the context of language modeling, this approach is challenging to apply because language data is finite-dimensional.

---

> > ### Comment · Reviewer_CUfq · 2024-08-07
> >
> > Questions have been addressed rigorously. Well done. Changing score from 6 -> 7.

---

### Official Review · Reviewer_JKEE · 2024-07-12

**Soundness:** 3
**Presentation:** 4
**Contribution:** 3
**Rating:** 7
**Confidence:** 3

**Summary:**

The authors present a new framework for unifying PDE surrogate modeling across domains. The proposed encoder, approximator, and decoder structure can accommodate PDEs of different discretizations and simulation types. Solutions are approximated in latent space, which aids in scalability and reducing computational cost of high-dimensional PDE data.

**Strengths:**

-	The method for encoding/decoding to a common latent space is exciting and seems to provide a good balance between compute/accuracy.
-	The experiments are done with challenging fluids problems, which provides a good evaluation of model capabilities.
-	The authors make a good effort to run a variety of experiments and provide good justification for when certain experiments cannot be performed. Furthermore, it is evident that the authors spent a lot of time thoroughly evaluating different model choices/hyperparameters and their effects.
-	The generalization study and latent space evaluation are convincing.
-	The authors consider Lagrangian fluid simulation, which is not commonly done in PDE surrogate modeling, and being able to accommodate this underexplored domain is good.
-	The authors provide a good background of the fluid data used and the clarity of the paper is good.

**Weaknesses:**

-	The Unet, FNO, and GINO benchmarks are good, however, they are not specifically tailored for efficient computation. There have been prior works that address latent space modeling or efficient PDE modeling of 3D data, namely Latent Spectral Models [1], Latent Evolution of PDEs (LE-PDE) [2], and FactFormer [3]. I would like to see some reasoning or experiments as to why your model is better.
-	The proposed architecture seems similar to OFormer [4]. It follows a similar encoding/propagating/decoding scheme as well as uses the transformer architecture. There are good improvements to adjust to higher dimensional and diverse data (e.g, Perceiver, supernodes, decoder design), but the broad architectural claims are shared with this previously proposed model (e.g. using a latent space, training an encoder/decoder, relying on transformer models for scalability). There are also other transformer-based works that present scalable architectures (DPOT [5], MPP [6]). I understand that these models do not include mechanisms for high-dimensional or irregular data, but I am curious to see how your model compares to these transformer-based models on a regular domain.
-	There is no justification given for why your model is a neural operator. I know that not every paper that proposes a neural operator justifies the operator approximation theorem, and there are good discretization invariance results, but I would like to see some reasoning for how your model satisfies operator approximation.
	Overall, I think the paper is a good contribution and is a valuable step in extending neural operators to large, complex PDE problems. There is a lack of comparable benchmarks, however this does not significantly detract from the experimental rigor and results.
1.	Haixu Wu, Tengge Hu, Huakun Luo, Jianmin Wang, Mingsheng Long, Solving High-Dimensional PDEs with Latent Spectral Models. https://arxiv.org/abs/2301.12664
2.	Tailin Wu, Takashi Maruyama, Jure Leskovec, Learning to Accelerate Partial Differential Equations via Latent Global Evolution, https://arxiv.org/abs/2206.07681
3.	Zijie Li, Dule Shu, Amir Barati Farimani, Scalable Transformer for PDE Surrogate Modeling, https://arxiv.org/abs/2305.17560
4.	Zijie Li, Kazem Meidani, Amir Barati Farimani, Transformer for Partial Differential Equations' Operator Learning, https://arxiv.org/abs/2205.13671
5.	Zhongkai Hao, Chang Su, Songming Liu, Julius Berner, Chengyang Ying, Hang Su, Anima Anandkumar, Jian Song, Jun Zhu, DPOT: Auto-Regressive Denoising Operator Transformer for Large-Scale PDE Pre-Training, https://arxiv.org/abs/2403.03542
6.	Michael McCabe, Bruno Régaldo-Saint Blancard, Liam Holden Parker, Ruben Ohana, Miles Cranmer, Alberto Bietti, Michael Eickenberg, Siavash Golkar, Geraud Krawezik, Francois Lanusse, Mariel Pettee, Tiberiu Tesileanu, Kyunghyun Cho, Shirley Ho, Multiple Physics Pretraining for Physical Surrogate Models, https://arxiv.org/abs/2310.02994

**Questions:**

-	I’m curious about the general behavior of stability of error accumulation of your model during auto-regressive rollout. Are there specific training details that you use to mitigate this? Does using a latent space help mitigate this effect?
-	What are your thoughts on using a latent space vs. directly working in physics space when it comes to the accuracy of the solutions? Latent unrolling is obviously faster during inference, but even in the physical domain many neural surrogates are still orders of magnitude faster than numerical solvers with the added benefit of being simpler and potentially more accurate than working in a latent space.

**Limitations:**

-	For common benchmark problems on a regular grid, I don’t see why this architecture would be preferred over a baseline or fine-tuning a pretrained model. Most physical problems use a mesh or irregular grid, but certain problems such as weather or isotropic turbulence come to mind.
-	Fixing the latent space is presented as a method to standardize different physics to a common space. However, more complex physics problems (3D turbulence vs. 2D laminar flows) may conceivably require larger latent spaces and the model would need to be retrained for different latent sizes.

---

> ### Author Rebuttal · Authors · 2024-08-07
>
> Thank you for your thorough review and helpful comments which were very useful to improve the paper. We address your points individually.
>
>
> **Why UPT is a better scalable latent space model**
>
> Latent Spectral Model uses geo-FNO (which could be consider as predecessor of GINO) to handle irregular grids. Therefore, it is somewhat similar to GINO as it has to map its input into a fixed regular grid representation. UPT doesn't map to a regular grid which makes it more flexible/expressive.
>
> LE-PDE only considers small-scale regular grid data and their design requires train and test data to have the *exact* same resolution (as mentioned in Appendix C of their paper). Therefore, its application to irregular grid data is impractical as a lot of data has different number of particles/mesh cells during test time. UPT can handle arbitrary input resolutions during training and generalizes well to higher input resolutions (as shown in Figure 5).
>
> FactFormer does not compress its input into a smaller latent space and therefore doesn't scale to large input sizes. Their method proposes an efficient attention mechanism, but their compute requirement would be similar to linear transformers and therefore become infeasible on large systems (as shown in Figure 2).
>
> **Comparison to transformer based models on regular grid**
>
> We compare UPT on two regular grid benchmarks as shown in the general response and in the supplemental pdf Tables 1 to 3. UPT outperforms all compared methods (such as OFormer or DPOT) -- often by quite a margin -- without being specifically designed for regular grid datasets.
>
>
>
> **Justification why UPTS are neural operators**
>
> Thank you for bringing up this important point. To avoid overloading the notation, we will provide a brief sketch of how universality can be established for our architecture. For transformer-based neural operators, universal approximation has been recently demonstrated in [1], Section 5. The arguments in this work are heavily based on [2], which establishes that nonlinearity and nonlocality are crucial for universality. By demonstrating that the attention mechanism can, under appropriate weight choices, function as an averaging operator, the results from [2] are directly applicable. For detailed proof, refer to Theorem 22 in [1].
> Our algorithm fits within this framework as well: we employ nonlinear, attention-based encoders and decoders (as allowed by the results of [2], Section 2) and utilize attention layers in the latent space.
>
> Please let us know if you require further explanations. We will provide detailed information in an updated version of the manuscript.
>
>
>
> [1] Calvello et al., "Continuum Attention for Neural Operators", arXiv 2024, https://arxiv.org/abs/2406.06486
>
> [2] Lanthaler et al., "The nonlocal neural operator: Universal approximation" arXiv 2024, https://arxiv.org/abs/2304.13221
>
>
> **Rollout stability**
>
> Our training procedure doesn't use any special methods to stabilize the rollout because this typically comes at increased training costs. However, one could easily apply such techniques to UPT.
> As we found our models to produce fairly stable rollouts even without additional techniques, we leave this direction for future work.
>
>
>
>
>
> **Thoughts on latent space vs physics space**
>
> We see the latent rollout as a promising way to speedup inference, particularly for large-scale systems where encoding and decoding takes up most of the runtime.
> UPT offers the flexibility to invest resources into training it as a latent rollout model (by using the inverse encoding/decoding objectives) or to save resources during training (by omitting the reconstruction objectives) at the cost of increased inference time. We consider this a valuable tool to have for neural operators.
>
> Additionally, the latent rollout enables applicability to Lagrangian simulations. As UPT models the underlying field instead of tracking individual particle positions it does not have access to particle locations at inference time. Therefore, autoregressive rollouts are impossible since the encoder requires particle positions as input. Using the latent rollout, it is sufficient to encode the initial particle positions into the latent space, which can then be propagated without the knowledge of any spatial positions. After propagating the latent space forward in time, one can simply query the latent space at arbitrary positions to evaluate the underlying field at given positions. We showcase this in Figure 7 where the latent space is queried with regular grid coordinates (white arrows).
>
>
>
> **Limitations of non-regular gridded data**
>
>
> We are interested in complex physics simulations which are often simulated using e.g., finite element meshes. Several phenomena are modeled by particle-based simulations such as smoothed particle hydrodynamics, material point methods, or discrete element methods. Many phenomena even require a coupling of different aforementioned simulation types. This is very much driven by daily life engineering applications, and as such was a main motivation for developing UPT.
>
>
>
>
>
> **Limitations on fixed latent space**
>
>
> In the current architecture of UPT, we consider a fixed latent space as it has proven to be an efficient way to compress the input into a fixed size representation to enable scaling to large-scale systems while remaining compute efficient.
> However, if an application requires a variable sized latent space, one could also remove the perceiver pooling layer in the encoder. With this change the number of supernodes is equal to the number of latent tokens and complex problems could be tackled by a larger supernode count. While we currently do not consider a setting where this is necessary, we show that the performance of UPT steadily increases with the number of supernodes during training (Figure 9 in Appendix C.4.3).
> Additionally, UPT's performance improves when more supernodes are used during evaluation than were utilized during training (Figure 8 in Appendix C.4.2).

---

> > ### Comment · Reviewer_JKEE · 2024-08-13
> >
> > Thanks for your detailed reply, most of my concerns have been addressed and I've raised the score. Best wishes!

---

### Official Review · Reviewer_5AUz · 2024-07-23

**Soundness:** 3
**Presentation:** 3
**Contribution:** 3
**Rating:** 6
**Confidence:** 4

**Summary:**

In this paper a framework for efficiently scaling neural operators is introduced under the name of Universal Physics Transformers (UPTs). It is a novel paradigm that scales neural operators across diverse spatio-temporal problems without relying on specific grid or particle-based structures. Leveraging transformer architectures, UPTs propagate dynamics within a compressed latent space, enabling efficient and flexible simulations.

**Strengths:**

1/ Originality:

The paper presents an original contribution to the field of neural operators by introducing Universal Physics Transformers (UPTs). This novel paradigm removes the traditional reliance on grid or particle-based latent structures, enabling greater flexibility and scalability across various simulation types. The innovative use of transformer architectures to propagate dynamics within a compressed latent space is a quite original combination of existing ideas applied to a new domain.

2/ Quality:

The quality of the overall contribution is good. The authors address a critical challenge in scaling neural operators for complex simulations and provide a robust framework with practical applications. The UPT framework is well-formulated, and the encoding and decoding schemes are designed to ensure efficient simulation rollouts. The experiments are comprehensive, covering multiple types of simulations, and the results demonstrate the superiority of UPTs in terms of performance and scalability.
Moreover, the paper's methodology is sound, and the technical claims are well-supported by evidence from the experiments.

3/ Clarity:

The paper is well-written and clearly presented. The authors provide a thorough background.

4/ Significance:

The significance of the paper lies in its potential impact on the broader scientific and engineering communities. By providing a unified and scalable framework for neural operators, UPTs can be applied to a wide range of spatio-temporal problems, including those in fluid dynamics. The ability to efficiently handle large-scale simulations can lead to significant advancements in these fields, offering valuable insights and solutions to complex physical phenomena.
Moreover, the experimental design is robust, with appropriate datasets and baselines used for comparison. The methodology is clearly described, allowing for reproducibility of the results. The use of inverse encoding and decoding techniques to enable latent rollouts is particularly noteworthy, as it demonstrates a deep understanding of the underlying principles and challenges in neural operator learning.

5/ Latente space and scalability:

The Universal Physics Transformers (UPTs) framework is efficient since it allows to get compact yet expressive latent space representation, capturing the essential dynamics of physical systems while significantly reducing memory usage and computational overhead. This compactness, combined with efficient encoding and decoding schemes, ensures that the transformation to and from the latent space is both efficient and accurate. By unifying the encoding of various grids and particles, UPTs simplify the modeling process and improve generalization across different spatio-temporal problems.

The empirical validation provided in the paper, through steady-state and transient flow simulations, demonstrates that UPTs achieve lower mean-squared error (MSE) and faster computation times compared to other models, highlighting the effectiveness of the latent space representation.

In terms of scalability, UPTs are designed to handle large-scale simulations effectively. The fixed-size latent space, regardless of input size, allows UPTs to manage large inputs without a proportional increase in computational cost. This scalability is evident in the framework's ability to maintain performance and efficiency with large meshes and numerous particles. The efficient latent space rollouts further enhance scalability by enabling quick updates and predictions, making UPTs suitable for time-dependent simulations. Additionally, the transformer architecture provides a robust foundation for scalability, ensuring that UPTs can handle complex spatio-temporal simulations efficiently.


6/ Lagrangian dynamic modeling:

The Universal Physics Transformers (UPTs) framework presents several notable strengths when viewed from the perspective of Lagrangian dynamic modeling. One of the primary advantages is the framework's ability to model particle-based simulations effectively without relying on traditional particle-structures. In Lagrangian simulations, particles move with the local deformation of the continuum, and modeling these dynamics accurately requires handling a large number of particles and their interactions.

UPTs use a unified latent space to encode particle information, enabling the framework to handle varying numbers of particles flexibly. This flexibility is particularly beneficial for Lagrangian methods, such as Smoothed Particle Hydrodynamics (SPH), where particle counts can vary significantly based on the simulation's complexity. By compressing this information into a fixed-size latent space, UPTs manage to reduce computational overhead while maintaining the ability to capture intricate particle interactions.

The efficient encoding and decoding processes in UPTs ensure that particle dynamics are propagated accurately and swiftly within the latent space. This efficiency is crucial for large-scale Lagrangian simulations, where the computational cost can be prohibitive with traditional methods. The ability to perform latent rollouts enables UPTs to predict future states quickly, making the framework suitable for real-time or near-real-time applications in Lagrangian dynamic modeling.

The paper provides strong empirical evidence of UPTs' effectiveness in Lagrangian dynamic modeling through experiments on datasets such as the Taylor-Green vortex in three dimensions (TGV3D). The results demonstrate that UPTs can effectively learn the underlying field dynamics and predict particle velocities with lower error compared to traditional Graph Neural Networks (GNNs) and other baselines. This empirical validation underscores the practical applicability of UPTs in complex Lagrangian simulations

UPTs exhibit strong generalization capabilities, which are critical for Lagrangian dynamic modeling. The ability to query the latent representation at any point in space-time allows UPTs to adapt to different particle distributions and simulation conditions without extensive retraining. This robustness ensures that UPTs can handle a wide range of Lagrangian problems, from small-scale particle systems to large-scale simulations involving a representative number of particles.

The use of transformers to manage the latent space representation in UPTs is a significant technical innovation. Transformers are known for their efficiency in handling large datasets and sequences, and their application in UPTs leverages this strength to manage the complexities of Lagrangian dynamics. This choice of architecture allows UPTs to model particle interactions accurately while maintaining computational efficiency.

**Weaknesses:**

1/ Clarity and detail in methodology:

The methodology, particularly the detailed implementation of the encoding and decoding schemes, could be more clearly articulated. Some parts of the algorithms are difficult to follow. Providing additional diagrams, detailed explanations, or pseudo-code would help clarify these complex processes. Ensuring that each step of the process is well-explained and easily understandable would make the paper more accessible and replicable.

2/ Scalability with extremely large datasets:

While the paper demonstrates UPTs' scalability, there is limited discussion on how the framework performs with extremely large datasets or in distributed computing environments. Given the increasing size of datasets in fields like high-resolution climate modeling and genomics, a discussion or preliminary results on the scalability of UPTs in such contexts would be valuable. This could include potential challenges, proposed solutions, and the impact on computational resources.

3/ Insufficient analysis of generalization capabilities:

While UPTs are shown to generalize across different simulation types, the paper lacks a deep analysis of this capability. Neural operator networks are often praised for their ability to generalize across different boundary conditions and initial states. A more thorough examination of UPTs' generalization performance, especially in unseen or out-of-distribution scenarios, would strengthen the claims. Detailed experiments and discussions on how UPTs perform under varied conditions would be valuable.

4/ Potential overfitting concerns:

Given the complexity and high capacity of transformer-based models, there is a risk of overfitting, especially on smaller datasets. The paper briefly mentions overfitting issues in some experiments but does not provide a detailed strategy for mitigating this. More discussion on regularization techniques, data augmentation strategies, or how UPTs handle overfitting would be beneficial. Insights into how the model can be generalized better and made more robust against overfitting are crucial for practical applications.

5/ Analysis from particle and grid based methods:

The Universal Physics Transformers (UPTs) framework offers a novel approach, but there are specific areas where it could be improved, particularly when compared to traditional particle and grid-based methods.

A/ Lack of detailed comparison with established methods:

The paper does not provide a thorough comparison with well-established particle-based methods (such as Smoothed Particle Hydrodynamics, SPH) and grid-based methods (such as Finite Volume Methods, FVM). Including detailed performance metrics, such as accuracy, computational cost, and scalability, would help highlight UPTs' advantages and areas needing improvement. Specifically, comparative studies on benchmark problems typically addressed by SPH (except figure 12) and FVM would provide more context.

B/ Handling of boundary conditions:

Particle and grid-based methods have well-established techniques for handling complex boundary conditions, which are often crucial in physical simulations. The paper does not sufficiently address how UPTs manage complex boundary conditions compared to these traditional methods. A more detailed discussion or experiments showcasing UPTs' effectiveness in dealing with various boundary conditions would strengthen the paper's claims.

C/ Adaptation to high-resolution grids and large particle systems:

While UPTs are shown to be scalable, the paper lacks detailed insights into their performance on extremely high-resolution grids or very large particle systems. Traditional methods often excel in these areas due to their specialized structures and optimizations. Providing more evidence on how UPTs handle such scenarios, including any potential bottlenecks and solutions, would be beneficial.

D/ Numerical stability and accuracy:

Grid-based methods, such as FVM, are known for their numerical stability and accuracy, especially in simulating fluid dynamics. The paper does not provide an in-depth analysis of UPTs' numerical stability and accuracy compared to these methods. Detailed experiments and discussions on how UPTs ensure stability and accuracy over long simulation times would enhance the paper’s credibility.

E/ Computational efficiency in complex geometries:

Particle and grid-based methods have specific strategies to efficiently handle complex geometries, such as adaptive meshing in grid-based methods or kernel adjustments in particle methods. The paper does not clearly explain how UPTs manage complex geometries and whether they can maintain computational efficiency in such scenarios. More detailed experiments or case studies involving complex geometrical domains would be informative.

F/ Interoperability with existing simulation tools:

Traditional methods are often integrated into comprehensive simulation tools (e.g., OpenFOAM for grid-based methods or LAMMPS for particle-based methods). The paper does not discuss how UPTs can be integrated or used alongside these existing tools, which is crucial for practical adoption. Providing insights into interoperability and how UPTs can complement or enhance traditional methods within established simulation frameworks would be useful.

G/  Handling of multiscale phenomena:

Particle and grid-based methods have developed sophisticated techniques to handle multiscale phenomena, such as adaptive mesh refinement (AMR) in grid-based methods. The paper lacks a discussion on how UPTs address multiscale phenomena, which are common in many physical simulations. Including experiments or theoretical discussions on UPTs' capabilities in multiscale modeling would be beneficial.


6/ Model conditioning standpoint:

Model conditioning is crucial for ensuring that neural networks adapt accurately to varying inputs and simulation conditions. The Universal Physics Transformers (UPTs) framework has room for improvement in this aspect. Here are specific areas where the paper could enhance its discussion and implementation of model conditioning.

A/ Insufficient detail on conditioning mechanisms:

The paper briefly mentions the use of feature modulation (e.g., DiT modulation) for conditioning UPTs to various inputs, such as the current timestep and boundary conditions. However, it lacks a detailed explanation of how these conditioning mechanisms are implemented and their impact on model performance. Providing more comprehensive descriptions and theoretical justifications for the chosen conditioning methods would help readers understand their effectiveness and potential limitations.

B/ Limited analysis of conditioning performance:

While the paper demonstrates that UPTs can handle different flow regimes and domains, it does not provide a thorough analysis of how well the conditioning mechanisms work across a broader range of scenarios. Including detailed experiments that specifically evaluate the performance of UPTs under different conditioning inputs, such as varying boundary conditions, initial states, and external forces, would strengthen the paper's claims.

C/ Scalability of conditioning methods:

The scalability of the conditioning mechanisms is not thoroughly discussed. As simulations grow in complexity, the effectiveness of conditioning methods can degrade if not properly scaled. The paper should address how the conditioning mechanisms scale with increasing input dimensions and simulation complexities, providing insights into any potential bottlenecks and how they are mitigated.

D/ Generalization to unseen conditions:

One of the strengths of neural operator networks is their ability to generalize to unseen conditions. The paper does not sufficiently explore how well UPTs generalize to entirely new boundary conditions or physical scenarios that were not present in the training data. Including experiments that test the generalization capabilities of UPTs to unseen conditions would provide valuable insights into the robustness of the conditioning methods.

E/  Comparison with other conditioning approaches:

The paper does not compare its conditioning mechanisms with other advanced conditioning techniques used in neural operator networks or related fields. A comparative analysis of different conditioning approaches, such as those used in Fourier Neural Operators and DeepONets, would highlight the strengths and weaknesses of the methods employed in UPTs.

F/  Impact of conditioning on training stability:

Conditioning mechanisms can significantly impact the training stability of neural networks. The paper does not discuss how the chosen conditioning methods affect the stability of UPT training, especially in the presence of complex and noisy data. Analyzing and addressing potential stability issues arising from conditioning would be beneficial.

G/ Real-world application scenarios:

While the paper presents conditioning in the context of fluid dynamics simulations, it does not discuss its applicability to other real-world scenarios that require complex conditioning. Exploring and providing evidence of UPTs' effectiveness in diverse applications, such as climate modeling or structural analysis, where conditioning to various external factors is crucial, would enhance the practical relevance of the framework.

**Questions:**

1/  The paper briefly mentions feature modulation for model conditioning but lacks detailed discussion on handling complex boundary conditions. Providing specific examples or additional experiments that illustrate UPTs' effectiveness in managing complex boundary conditions would strengthen the paper’s claims. Addressing this can clarify the practical applicability of UPTs in real-world scenarios where boundary conditions play a crucial role.

How does the UPT framework handle complex boundary conditions compared to traditional particle and grid-based methods?

2/  While UPTs are presented as efficient and scalable, the paper does not delve into the specifics of their numerical stability and accuracy compared to traditional methods. Discussing strategies to maintain stability and accuracy over extended simulations and providing relevant experimental evidence would address potential concerns about the reliability of UPTs in long-term applications.

What measures are in place to ensure the numerical stability and accuracy of UPTs over long simulation periods?

3/ The paper demonstrates scalability and generalization within certain limits but does not extensively explore these aspects in more demanding scenarios. Providing additional experimental results or theoretical insights on UPTs’ performance in high-resolution and large-scale environments, as well as their ability to generalize to entirely new conditions, would significantly bolster the paper’s contribution and practical relevance.

 What are the scalability and generalization capabilities of UPTs when applied to extremely high-resolution grids, large particle systems, and unseen physical scenarios?

**Limitations:**

Yes.

---

> ### Author Rebuttal · Authors · 2024-08-07
>
> Thank you for your profound review and suggestions that helped us to improve our paper a lot. We addressed all your comments and followed all your suggestions. Please let us expand a bit on your comments.
>
>
> **Clarity and detail in methodology**
>
>
> In order to make the paper more accessible we -- as suggested -- add pseudocode of a UPT forward pass with adaptive sampling. This together with sketch Figure 3a, sketch Figure 3b, and the encoder/approximator/decoder description of Section 3 should make the paper understandable wrt. all necessary details.
>
> **Scalability to extremely large datasets / high-resolution grids**
>
> We agree that scalability to extremely large datasets is a desired property. In the paper, we have done our best to test this. Our models are trained using distributed training up to 32 A100 GPUs. We have self-simulated a dedicated dataset which is larger (> 50000 mesh points), and more complex (different flow regimes, different number of differently placed obstacles) as standard grid based datasets. As requested by several reviewers we have added experiments on the Shallow Water dataset and the Navier-Stokes dataset to the paper, which demonstrate the diverse applicability of UPT. However, even larger simulations than our transient flow simulations are beyond the scope of this paper and the available compute budget. Nevertheless, we have tested GPU usage of several model classes (Figure 2), which demonstrates that our framework is able to process meshes / particle clouds with up to 4 million nodes on a single GPU.
>
> **Insufficient analysis of generalization capabilities**
>
> We agree that generalization capabilities are one of the most important aspects to test for neural operators. Therefore, in the transient flow experiment -- our largest experiment -- we test generalization across number of input / output points (discretization convergence), discretization across different flow regimes (input velocity), and generalization across different scenarios (differently placed obstacles).
>
>
> **Potential overfitting concerns**
>
> We observe that the variable selection of the supernodes and the variable selection of input/output nodes allows for a strong regularization and data augmentation. The general training pipeline of UPT is therefore very robust against overfitting. This is further strengthened by the additional Navier-Stokes and Shallow water experiments.
>
> **Lack of detailed comparison with established numerical methods**
>
> We are not claiming to be better than numerical methods. We compare with neural methods and especially focus on scaling. We follow the general procedures of the community, and report MSE, correlation time measures, and runtime comparisons. The latter two give an idea of the potential of UPT when compared with numerical methods. Similar to e.g. weather modeling (see Aurora [2], Pangu [3], ...) we follow the belief that benefits of neural operators will become more pronounced at scale.
>
> **Handling of boundary conditions**
>
> The ShapeNet car experiment allows for testing neural operators on complicated geometries. UPT performs favorably, even when using a strongly reduced latent space representation. Additionally, in the transient flow example we have varying numbers of differently placed obstacles which can also be seen as complicated in-domain boundaries. In fact, UPT is specifically designed for large data on non-regular domains including different boundary conditions.
>
> **Model conditioning**
>
> We have added a more detailed paragraph on the model conditioning choices in this paper. In our paper we follow closely the detailed studies described in [1].
>
> **Real world applications**
>
> Real world applications are beyond the scope of this work, but a very strong motivation for developing our framework. We note that current weather modeling such as Aurora [2] or Pangu [3] follow a similar design principle, but for regular gridded data, and without discretization convergence properties.
>
> **Question on complex boundary conditions**
>
> Transformers offer a flexible way to encode various boundary conditions.
> Skalar boundary conditions, such as inflow velocity or the current timestep, can be encoded via feature modulations. We use DiT modulation as it has shown strong performances in transformers.
>
> Additionally, transformers offer a flexible mechanism to encode additional information by encoding the information into tokens and concatenating them to the supernodes or latent tokens. We make use of this flexible encoding in the ShapeNet-Car experiments where we additionally encode the signed distance function evaluated on a 3D grid via a CNN and feed the resulting tokens to the approximator.
>
> **Question on numerical stabilitiy and accuracy**
>
> Traditional methods require extremely small timesteps in order to preserve numerical stability when resolving the physics.
> As neural operators only approximate the solution of the traditional solver, they can operate on much larger timesteps as their powerful modeling capabilities enables them to learn dynamics also from a coarse time resolution.
>
>
> Furthermore, UPT rollouts are fairly stable without any specific measures to ensure rollout stability. Sophisticated techniques to improve rollout stability (e.g., [4]) could be easily applied to UPTs to further enhance rollout stability. However, as these techniques typically impose a runtime overhead, we leave exploration thereof to future work.
>
>
> [1] Gupta et al., "Towards Multi-spatiotemporal-scale Generalized PDE Modeling", arXiv 2022, https://arxiv.org/abs/2209.15616
>
> [2] Bodnar et al., "Aurora: A foundation model of the atmosphere", arXiv 2024, https://arxiv.org/abs/2405.13063
>
> [3] Bi et al., "Accurate medium-range global weather forecasting with 3D neural networks", Nature 2023, https://www.nature.com/articles/s41586-023-06185-3
>
>
> [4] Brandstetter et al., "Message passing neural pde solvers", ICLR 2022, https://arxiv.org/abs/2209.15616

---

> > ### Comment · Reviewer_5AUz · 2024-08-09
> > **Summary of the revision and follow-up**
> >
> > Thank you for addressing the feedback provided in my initial review. Here is a summary of the revisions and responses you have made, along with some additional feedback and observations.
> >
> > **Scalability to extremely large datasets / high-resolution grids**
> >
> > I commend your efforts to demonstrate scalability within the constraints of your computational resources. The additional experiments on the Shallow Water and Navier-Stokes datasets effectively illustrate your framework's ability to handle large datasets. Your discussion on using distributed training across multiple GPUs highlights the potential of your approach in handling complex simulations.
> >
> > **Generalization capabilities**
> >
> > The expanded tests on generalization, including across different flow regimes and discretizations, strengthen the evidence for your framework's robustness. This comprehensive analysis is a significant enhancement and effectively addresses previous concerns about the adaptability of your model.
> >
> > **Handling of boundary conditions**
> >
> > The examples provided on handling complex boundary conditions through experiments like the ShapeNet car and transient flow simulations highlight the flexibility and adaptability of UPTs in various scenarios. This aspect of your work is well-explained and a notable strength of your framework.

---

> > ### Comment · Reviewer_5AUz · 2024-08-09
> > **Suggestions**
> >
> > 1. While real-world applications are beyond the current scope, discussing potential use cases or hypothetical scenarios could highlight the practical impact of UPTs.
> > 2. Consider exploring how UPTs might be integrated with traditional particle or grid-based methods in practice. Discussing potential hybrid approaches could broaden the applicability of your work
> > 3. Discussing specific measures or techniques to enhance long-term stability and robustness in simulations, particularly with noisy data, could strengthen the paper's claims.
> > 4. As hardware capabilities evolve, exploring how UPTs can leverage advancements in GPU and TPU technologies could provide insights into the future potential of your framework.

---

> > ### Comment · Reviewer_5AUz · 2024-08-09
> > **Follow-up questions**
> >
> > 1. In your response, you mentioned enhancing the model conditioning with techniques such as feature modulation, but I could not find the additional details in the paper. Could you elaborate on how the DiT modulation and any other conditioning mechanisms are implemented in the UPT framework, and how these techniques specifically improve model adaptability to varying boundary conditions and simulation inputs?
> > 2. You highlighted the use of transformers to manage complex boundary conditions, particularly in experiments like the ShapeNet car. Could you provide more details on how the transformers are configured to encode these boundary conditions effectively? Are there specific architectural adjustments or tokenization strategies employed to ensure accurate representation and processing of boundary conditions?
> > 3. In the context of your framework's numerical stability, you mentioned that UPT rollouts are stable without specific measures. Could you clarify how the UPT framework maintains stability and accuracy over extended simulations, especially when compared to traditional numerical solvers that require small timesteps for stability? Are there specific aspects of the transformer architecture that contribute to this stability?
> > 4. While you provided insights into UPT's scalability with large datasets and distributed training, can you discuss any potential limitations or bottlenecks encountered when scaling the model further? How does the framework handle challenges such as increased computational demands or memory usage as the complexity of the simulations and datasets grows?

---

> > ### Comment · Reviewer_5AUz · 2024-08-09
> > **Missing pseudo-code and extra details on model conditioning**
> >
> > I am unable to find the pseudo-code and the additional details on model conditioning that are mentioned in your response.

---

> > > ### Author Response · Authors · 2024-08-09
> > > **Pseudo-code for training UPT**
> > >
> > > Due to space constraint in the rebuttal, we were not able to include it there (the rebuttal did also not allow to update the submitted paper). Please find below pseudocode for UPT in the setting of our transient flow experiments (2D positions with 3 features per node).
> > >
> > > ```
> > > # input_embed: linear projection, projecting from the number of input features to a hidden dimension
> > > # pos_embed: positional embedding (with sine and cosine waves of different frequencies) as common in transformers
> > > # message_mlp: shallow MLP to create messages
> > > # encoder_transformer: stack of transformer blocks of the encoder
> > > # latent_queries: `n_latent_tokens` learnable query vectors
> > > # encoder_perceiver: cross attention block
> > > # approximator_transformer: stack of transformer blocks of the approximator
> > > # query_mlp: shallow MLP in the decoder to encode query positions
> > > # decoder: cross attention block
> > >
> > > def encoder(input_features, input_pos, radius, n_supernodes):
> > >     """
> > >     encode arbitrary pointclouds into a fixed latent space
> > >     inputs:
> > >         input_features `Tensor(n_input_nodes, 3)`: features of input nodes
> > >         input_pos `Tensor(n_input_nodes, 2)`: positions of input nodes
> > >         n_supernodes `integer`: number of supernodes
> > >         radius `float`: radius for creating the radius_graph
> > >     outputs:
> > >         latent `Tensor(n_latent_tokens, hidden_dim)`: encoded latent space
> > >     """
> > >     # create radius graph (using all input nodes)
> > >     # edges are uni-directional and are passed from nodes_from to nodes_to
> > >     nodes_from, nodes_to = radius_graph(input_pos, radius)
> > >
> > >     # select supernodes from input_nodes
> > >     n_input_nodes = len(input_features)
> > >     supernode_idx = randperm(n_input_nodes)[:n_supernodes]
> > >
> > >     # filter out edges that do not involve supernodes
> > >     is_supernode_edge = nodes_to in supernode_idx
> > >     nodes_from = nodes_from[is_supernode_edge]
> > >     nodes_to = nodes_to[is_supernode_edge]
> > >
> > >     # encode inputs and positions
> > >     encoded_nodes = input_embed(input_features) + pos_embed(input_pos)
> > >
> > >     # create messages
> > >     messages = message_mlp(encoded_nodes[nodes_from])
> > >     # accumulate messages per supernode by averaging messages
> > >     supernodes = accumulate_messages(messages, nodes_to, reduce="mean")
> > >
> > >     # process supernodes with some transformer blocks
> > >     supernodes = encoder_transformer(supernodes)
> > >     # perceiver pooling from supernodes to latent tokens
> > >     latent = encoder_perceiver(query=latent_queryies, key=supernodes, value=supernodes)
> > >     return latent
> > >
> > > def approximator(latent):
> > >     """
> > >     propagates latent forward in time
> > >     inputs:
> > >         latent_t `Tensor(n_latent_tokens, hidden_dim)`: encoded latent space at timestep t
> > >     outputs:
> > >         latent_t_plus_1 `Tensor(n_latent_tokens, hidden_dim)`: encoded latent space at timestep t + 1
> > >     """
> > >     return approximator_transformer(latent)
> > >
> > > def decoder(latent, query_pos):
> > >     """
> > >     decode latent space pointwise at arbitrary positions
> > >     inputs:
> > >         latent `Tensor(n_latent_tokens, hidden_dim)`:
> > >         query_pos `Tensor(n_outputs, 2)`: positions for querying the latent space
> > >     outputs:
> > >         decoded `Tensor(n_outputs, 3)`: evaluation of the latent space at query positions
> > >     """
> > >     # encode query positions
> > >     query_pos_embed = query_mlp(pos_embed(query_pos))
> > >     # query latent space
> > >     decoded = decoder(query=query_pos_embed, key=latent, value=latent)
> > >     return decoded
> > >
> > >
> > > def train_step(input_features, input_pos, n_supernodes, radius, query_pos):
> > >     """
> > >     encode arbitrary pointclouds into a fixed latent space
> > >     inputs:
> > >         input_features `Tensor(n_input_nodes, 3)`: features of input nodes at timestep t
> > >         input_pos `Tensor(n_input_nodes, 2)`: positions of input nodes at timestep t
> > >         n_supernodes `integer`: number of supernodes
> > >         radius `float`: radius for creating the radius_graph
> > >         query_pos `Tensor(n_outputs, 2)`: positions for querying the latent space
> > >         target_features `Tensor(n_input_nodes, 3)`: features of nodes at timestep t + 1
> > >     outputs:
> > >         loss `Tensor`: skalar loss value
> > >     """
> > >     # next-step prediction
> > >     latent_t = encoder(input_features, input_pos, radius, n_supernodes)
> > >     latent_tplus1 = approximator(latent_t)
> > >     decoded_tplus1 = decoder(latent_tplus1, query_pos)
> > >     next_step_loss = mse(decoded_tplus1, target_features)
> > >
> > >     # inverse decoder (decode latent into inputs)
> > >     decoded_t = decoder(latent_t, input_pos)
> > >     inverse_decoding_loss = mse(decoded_t, input_features)
> > >
> > >     # inverse encoder (encode predictions at t + 1 into latent of t + 1)
> > >     inverse_encoded = encoder(decoded_tplus1, query_pos, radius, n_supernodes)
> > >     inverse_encoding_loss = mse(inverse_encoded, latent_tplus1)
> > >
> > >     return next_step_loss + inverse_decoding_loss + inverse_encoding_loss
> > > ```

---

> > > > ### Comment · Reviewer_5AUz · 2024-08-13
> > > > **Thank you for the follow-up**
> > > >
> > > > Please add the discussion in the revised version.

---

> ### Author Response · Authors · 2024-08-09
> **Reply to follow-up questions**
>
> We updated the paper with additional details concerning modulation which would be included in the camera-ready version, but the rebuttal unfortunately did not allow updating the paper.
>
> **1. Details on modulation**
>
> We use the DiT modulation as it was introduced in the original paper [1] and condition onto boundary conditions (timestep, inflow velocity) in the following way:
>
> ```
> # embed: embedding layer (with sine and cosine waves of different frequencies) as also common to encode positions in transformers
> # condition_mlp: shallow MLP to combine boundary conditions
> def create_condition(timestep, velocity):
>     timestep_embed = embed(timestep)
>     velocity_embed = embed(velocity)
>     embed = concat([timestep_embed, velocity_embed])
>     condition = condition_mlp(embed)
>     return condition
> ```
>
> From this condition, DiT modulates its features by (i) shifting the features after layer normalization (ii) scaling the features after layer normalization or (iii) gating the features before adding the residual.
> This is done in the same way as DiT does it. We provide a link to the original implementation [2] but are also happy to provide pseudocode if needed.
>
>
> This form of conditioning allows us to encode the inflow velocity and the current timestep into the model. Therefore, the model can better adapt to different inflow velocities as this information can be explicitly encoded into the model. Without encoding such boundary conditions into the model, it would need to derive it from the mesh points or particles alone, which is sometimes impossible.
> Note that we also apply modulation to UNets in the form of FiLM [3] (similar to DiT it scales and shifts intermediate features) and for FNO/GINO we apply the "spatial-spectral parameter conditioning" from [4] (see Appendix B.2.6) which applies modulation in the fourier space of FNO.
>
>
> [1] Peebles et al., "Scalable Diffusion Models with Transformers", ICCV 2023, https://arxiv.org/abs/2212.09748
>
> [2] https://github.com/facebookresearch/DiT/blob/main/models.py#L101
>
> [3] Perez et al., "FiLM: Visual Reasoning with a General Conditioning Layer" AAAI 2018, https://arxiv.org/abs/1709.07871
>
> [4] Gupta et al., "Towards Multi-spatiotemporal-scale Generalized PDE Modeling", arXiv 2022, https://arxiv.org/abs/2209.15616
>
>
> **2. Boundary conditions in ShapeNet-Car**
>
> In the ShapeNet-Car experiments, we showcase the flexible encoding of UPT by additionally encoding the signed distance function (SDF) of the geometry.
> Given SDF evaluated on a 3D grid, we encode it with a shallow CNN with 3D convolutions. For example, we use a 64^3 grid of SDF values and encode it into a 8^3 latent representation with said CNN.
> This grid is then flattened out into a 1D sequence of 8^3=512 tokens. Afterwards these 512 tokens are concatenated to the output of the encoder (the 1024 latent tokens from encoding the mesh of the car) resulting in 1536 tokens as input to the approximator.
>
> **3. Numerical stability of UPT**
>
> Traditional solvers resolve the physics in very small timesteps as this is required to preserve the physical properties of the simulation. For example, if two particles are about to collide but the time resolution is not small enough to accurately detect when the two particles collide (and therefore change direction/velocity), the two particles will overlap with each other in the next simulation timestep. If this overlap is large, the resulting force acting on these particles will be extremely high, which in turn results in numerical instability and a collapse of the simulation.
> Contrary, UPTs do not learn to explicitly model the interactions between two particles, but rather an abstract representation thereof. Therefore, UPT does not need to resolve at a fine-grained time resolution as it can learn that "if two particles are about to collide, they are about to change direction". This makes it much more resilient against numerical stability as it focuses on abstract concepts instead of low-level interactions.
>
>
> Other neural operators like GNNs or graph neural operators (GNOs) also resolve particle-particle interactions and therefore require smaller timesteps than UPT. We showcase this in Figure 6 of our paper, where the UPT modeling paradigm (via an abstract compressed latent space) can resolve much larger timesteps without problems. We show this in our experiments on TGV3D where UPT is trained on 10x coarser timesteps than GNS or SEGNN.
>
>
> **4. Scaling properties of UPTs**
>
> As UPTs consist of mainly transformer layers, it does not have any issues with scaling. The same techniques that are applied to train large language models (LLMs) can be applied to UPTs as well. As an example, Llama3-405B [1] was scaled to 16K H100 GPUs. While we do not envision that we train UPT on such a massive scale in the foreseeable future, the used techniques can be readily applied to UPTs.
>
> [1] Dubey et al., "The Llama 3 Herd of Models", arXiv 2024, https://arxiv.org/abs/2407.21783

---

### Author Rebuttal · Authors · 2024-08-07

We thank all reviewers for their positive feedback and for their constructive comments and suggestions.
We are pleased to see that the reviewers highlighted the clarity of our paper and appreciated our detailed and thorough experiments. Several reviewers recognized the efficiency of our approach by encoding and decoding into a latent space as well as UPTs ability to handle both Lagrangian and Eulerian data.



To address the questions of the reviewers, we included the following new experiments. Please find visualizations thereof in the supplemental rebuttal pdf.

**Experiments on a regular grid Navier-Stokes equations dataset**

We run comparisons against different Transformer baselines on regular gridded Navier-Stokes equations data (data, baseline results and evaluation protocol taken from [1]). UPT outperforms all compared methods, some of which are specifically designed for regularly gridded data.

As baseline transformers often train small models, we first compare on a small scale, where UPT significantly outperforms other models.

| Model | # Params | Rel. L2 Error  |
|---|---|---|
| FNO | 0.5M | 9.12 \%  |
| FFNO | 1.3M | 8.39 \%  |
| GK-T |  1.6M | 9.52 \%  |
| GNOT |  1.8M | 17.20 \%  |
| OFormer | 1.9M |  13.50 \%  |
| UPT-T | 1.8M | **5.08** \%  |

We also compare on larger scales, where UPT again outperforms competitors, even if they train much larger models or pre-train (PT) on more data followed by fine-tuning (FT) on the Navier Stokes dataset.


| Model | # Params | Rel. L2 Error  |
|---|---|---|
| DPOT-Ti |  7M | 12.50 \%  |
| DPOT-S | 30M |  9.91 \%  |
| DPOT-L (PT) | 500M | 7.98 \%  |
| DPOT-L (FT) | 500M | 2.78 \%  |
| DPOT-H (PT) | 1.03B | 3.79 \%  |
| CViT-S | 13M | 3.75 \%  |
| CViT-B | 30M | 3.18 \%  |
| UPT-S | 13M | 3.12 \%  |
| UPT-B | 30M | **2.69** \%  |


[1] Hao et al., "DPOT: Auto-Regressive Denoising Operator Transformer for Large-Scale PDE Pre-Training", ICML 2024, https://arxiv.org/abs/2403.03542



**Experiments on a regular grid Shallow Water equations dataset**


We run comparisons against UNet, FNO, Dilated ResNet variants on regular gridded Shallow Water eqations data (data, baseline results and evaluation protocol taken from [2]). Similarly to the Navier-Stokes experiments, UPT can outperform methods which are specifically designed for regularly gridded data.


| Model | # Params | Rel. L2 Error  |
|---|---|---|
| DilResNet | 4.2M | 13.20 \%  |
| U-Net | 148M | 5.68 \%  |
| FNO | 268M | 3.97 \%  |
| CViT-S | 13M | 4.47 \%  |
| UPT-S | 13M | **3.96** \%  |

[2] Wang et al., "Bridging Operator Learning and Neural Fields:. A Unifying Perspective", arXiv 2024, https://arxiv.org/abs/2405.13998

**Impact of different positional encodings**

We added ablation results with different positional encoding.
Using a Fourier feature mapping from [3] resulted in an slightly better performance in the TGV2D experiments. We visualize this in Figure 1 of the supplemental rebuttal pdf.

[3] Tancik et al., "Fourier features let networks learn high frequency functions in low dimensional domains", NeurIPS 2020, https://arxiv.org/abs/2006.10739


**UPT scaling clarifications**

We added experiments to clarify the scaling of UPTs in the transient flow experiments by training small UPT models that perform similar to larger GINO models. We visualize the results in Figure 2 of the supplemental rebuttal pdf, which shows that UPT scales well when increasing parameter counts. This experiment should clarify that UPTs do scale well with parameter counts, but as the testloss of UPT (in Figure 5 of the paper) is significantly lower than for GINO, it is much harder to improve it further.


**Scaling with dataset size**

We investigate scaling behavior of UPT w.r.t. dataset size by training a UPT-8M model on smaller subsets in the setting of our transient flow experiments. Figure 3 in the supplementary rebuttal pdf visualizes the scaling curves. UPT scales well with data and achieves comparable results to GINO-8M with 4x less data.


**Further additions to the paper**

We followed the reviewers comments and suggestions, e.g., by including pseudocode of an UPT forward pass, detailed paragraphs on positional encoding and conditioning mechanisms, implementation details of the supernode pooling/radius graph and additional related works.


In addition, we followed the reviewers comments and suggestions, e.g. by discussing their raised points in the Discussion section or added clarifications, contextualization, and references.

---

### Decision · Program_Chairs · 2024-09-25

**Decision:**

Accept (poster)

**Comment:**

This paper studies an encoder-decoder arch of Transformers as neural operators for several PDE tasks, mainly flow problems. The authors had nice discussions with the reviewers. However, after reading the paper, I felt quite unsatisfied with the method presented, especially with regard to the scaling properties. The scaling law of Transformer considers the increasing "complexity" of the dataset. Yet, here in this paper, the models are still trained separately for each dataset (4.1-4.3), and the study of scaling is with respect to the size of discretizations (# of super nodes). Note that this "fact", even though ad-hoc, is quite commonly acknowledged in the CFD community from the work of Temam, Shen, Tadmor, and many others: when isotropic turbulence formed (so that one can observe the energy cascade), the nonlinearity is quite "simple" and has a lower intrinsic dimension (the reason reduced-basis approach works for this regime) if the initial conditions are sampled from an energy distribution with certain decay. In this regard, unlike language, the "complexity" of the dataset does not increase with the increase of the tokens. Another note is that OpenFOAM's solver prioritizes speed over the conformity to the NSE in the continuous Hilbertian framework, e.g., RANS already makes things tamer than DNS. Nevertheless, the architecture is simple and may have long lasting impact on the operator learning community.